# CH-RUN: A deep-learning-based spatially contiguous runoff reconstruction for Switzerland

Basil Kraft[1], Michael Schirmer[2], William H. Aeberhard[3], Massimiliano Zappa[2], Sonia I. Seneviratne[1], and Lukas Gudmundsson[1]

[1]Institute for Atmospheric and Climate Science (IAC), ETH, Zurich, Switzerland
[2]Swiss Federal Research Institute (WSL), Birmensdorf, Switzerland
[3]Swiss Data Science Center, ETH, Zurich, Switzerland

**Correspondence:** Basil Kraft (basil.kraft@env.ethz.ch)

**Abstract.**

This study presents a data-driven reconstruction of daily runoff that covers the entirety of Switzerland over an extensive period from 1962 to 2023. To this end, we harness the capabilities of deep learning-based models to learn complex runoff-generating processes directly from observations, thereby facilitating efficient large-scale simulation of runoff rates at ungauged locations. We test two sequential deep learning architectures: a long short-term memory (LSTM) model, a recurrent neural network able to learn complex temporal features from sequences, and a convolution-based model, which learns temporal dependencies via 1D convolutions in the time domain. The models receive temperature, precipitation, and static catchment properties as input. By driving the resulting model with gridded temperature and precipitation data available since the 1960s, we provide a spatiotemporally continuous reconstruction of runoff. The efficacy of the developed model is thoroughly assessed through spatiotemporal cross-validation and compared against a distributed hydrological model used operationally in Switzerland.

The developed data-driven model demonstrates not only competitive performance but also notable improvements over traditional hydrological modeling in replicating daily runoff patterns, capturing interannual variability, and discerning long-term trends. The resulting long-term reconstruction of runoff is subsequently used to delineate substantial shifts in Swiss water resources throughout the past decades. These are characterized by an increased occurrence of dry years, contributing to a negative decadal trend in runoff, particularly during the summer months. These insights are pivotal for the understanding and management of water resources, particularly in the context of climate change and environmental conservation. The reconstruction product is made available online.

Furthermore, the low data requirements and computational efficiency of our model pave the way for simulating diverse scenarios and conducting comprehensive climate attribution studies. This represents a substantial progression in the field, allowing for the analysis of thousands of scenarios in a time frame significantly shorter than traditional methods.

# 1 Introduction

Hydrological modeling and runoff prediction are critical for understanding and managing water resources, particularly in the face of climate change and increasing human impacts on the environment (Seneviratne et al., 2021; Arias et al., 2023). In Switzerland, a country characterized by diverse topography and climatic conditions, understanding and predicting runoff patterns is essential for effective water management, flood control, and environmental conservation (Brunner et al., 2019a).

Traditional hydrological models offer pivotal insights into land-surface processes. For Switzerland, a diverse array of hydrological models has been employed (Horton et al., 2022), ranging from complex ones, which are heavily founded on physical principles, to lightweight ones using conceptual process representations with calibrated parameters. While the former offer detailed insights and control, they rely on a large number of inputs and are computationally expensive. The latter, in contrast, can be parsimonious in terms of data and computational resources, yet they need to be calibrated per catchment, which limits their applicability to prediction in ungauged catchments. The generalization to ungauged catchments via regionalization is possible, but introduces another layer of complexity (Beck et al., 2016). As a complementary approach, deep learning holds potential as a tool for hydrological modeling, both in terms of performance and efficiency (Nearing et al., 2021), and it comes with built-in regionalization when trained on multiple catchments jointly (Kratzert et al., 2024).

The potential of machine learning to represent land surface processes, including runoff, has been widely demonstrated and discussed (Camps-Valls et al., 2021; Reichstein et al., 2018; Kraft et al., 2019; Gudmundsson and Seneviratne, 2015; Ghiggi et al., 2021). Deep learning, in particular, has shown promise for nowcasting and forecasting runoff at gauged catchments, aiding in warning systems for extreme flow events (Kratzert et al., 2018; Gauch et al., 2021a). It is, however, less common to employ data-driven models for large-scale reconstruction and monitoring (Nasreen et al., 2022). Reconstruction products are widely used for process understanding, investigation of long-term trends, and study of extreme events within a wider spatial and temporal context (Gudmundsson and Seneviratne, 2015; Ghiggi et al., 2019, 2021; Muelchi et al., 2022). In addition, machine learning-based models enable simulation of scenarios and real-time monitoring with significant speedup (Reichstein et al., 2019; Kraft et al., 2021).

This study introduces a data-driven approach to reconstructing daily runoff in Switzerland with contiguous spatial coverage, spanning an extensive period from 1962 to 2023 with the potential for continuous updates. We optimize a range of neural network-based models in different setups and evaluate their performance at the catchment-level in a comprehensive spatiotemporal cross-validation scheme. The results are benchmarked against simulations from the PREVAH hydrological model, which is used operationally in Switzerland. The extended coverage compared to PREVAH is enabled by reduced data requirements by only using temperature and precipitation as meteorological drivers. In the Swiss context, these variables are available as regular grids since the 1960s, while additional variables, such as relative humidity and wind speed, are available starting in the 1980s. The best-performing model is subsequently used to reconstruct daily runoff rates with complete spatial and temporal coverage since the 1960s. The paper closes with a discussion of the strengths and limitations of the approach and first insights from the extended reconstruction.

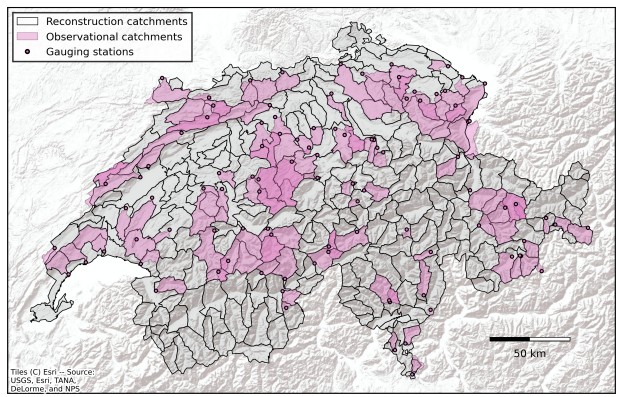

**Figure 1.** From sparse observations with low anthropogenic impact to contiguous spatial coverage. The 98 observational catchments highlighted in magenta were selected to only be marginally affected by anthropogenic factors and served as a base for training and evaluating the data-driven models. These catchments are of similar size as the target catchments for reconstruction (gray).

## 2   Data

### 2.1   Runoff observations

The observed discharges were taken from the CAMELS-CH dataset (Höge et al., 2023), which was updated with current data from the Swiss Federal Office for the Environment (FOEN, 2024) and supplemented by stations operated by the cantons Aargau, Baselland, Bern, St. Gallen and Zurich. In total, there were 267 stations available. A subset of 98 catchments was selected to minimize the anthropogenic impact (Fig. 1), i.e., no hydropower plant or reservoir was located upstream of the gauging station. This selection was based on the attributes of the CAMELS-CH dataset and insights from a previous study (Brunner et al., 2019c).

### 2.2   Meteorological drivers

We considered daily precipitation and air temperature as meteorological drivers, from interpolated observational data with a spatial resolution of 1 km, namely the daily gridded datasets RhiresD (Schwarb, 2000; MeteoSwiss, 2021a) and TabsD (Frei, 2014; MeteoSwiss, 2021b). Gridded daily temperature and precipitation were spatially averaged for all considered catchments.

### 2.3   Catchment properties

For the across-catchment modeling of runoff, a set of static catchment properties was considered. These variables can improve generalization to catchments not seen during training. The static variables used are identical to those needed for forcing the spatially distributed PREVAH model (see next section) and include elevation, aspect, land use, soil depth, soil water holding capacity, hydraulic conductivity and two further indices describing soil and hydraulic properties. These gridded variables were

aggregated to appropriate catchment values depending on the level of measurement, e.g., a circular mean for aspect or a distribution of classes within each catchment for land use. As compared to previous applications of PREVAH (Viviroli et al., 2009b; Speich et al., 2015), the sources of the data providing the static properties have been updated and include:

- The swissALTI3D digital elevation model (swisstopo, 2018; Weidmann et al., 2018),

- the new habitat map of Switzerland (Price et al., 2023), aggregated to match the land use classes integrated in PREVAH,

- the Soilgrids products (Hengl et al., 2017; Poggio et al., 2021), enriched with high-resolution data for Swiss forests (Baltensweiler et al., 2022), merged and used to estimate soil properties, and

- the recent information on extent of glaciers in Switzerland (Linsbauer et al., 2021).

## 2.4 PREVAH runoff simulations as benchmark

The modelling of Swiss catchments has a long history in hydrology research (Horton et al., 2022; Addor and Melsen, 2019). Among a set of 21 models compared, Horton et al. (2022) found the PREVAH (PREcipitation-Runoff-EVApotranspiration Hydrological response unit model) model (Viviroli et al., 2009b) to be the most commonly used model with applications going from the plot scale process evaluation (Zappa and Gurtz, 2003), to the operational implementation for drought anticipation
(Bogner et al., 2022), and to the Switzerland-wide assessment of climate impacts on hydrology (Brunner et al., 2019a). Furthermore, a PREVAH-based baseline is included in the Swiss version of the CAMELS (Höge et al., 2023) data set (catchment attributes and meteorology for large-sample studies) as introduced by Addor et al. (2017).

For this study, we created a benchmark runoff simulation for the selected catchments on the basis of PREVAH. The simulations cover the period from 1981 until the end of 2022. The procedure adopted to obtain the PREVAH-benchmark closely
follows the methodologies presented in previous studies (Speich et al., 2015; Brunner et al., 2019c; Höge et al., 2023). The gridded version of PREVAH (Viviroli et al., 2009b; Speich et al., 2015) has been applied at 500 m resolution. The time series of the investigated catchments were then obtained by spatially averaging daily gridded values. For further details on the setup and application of PREVAH, we refer to the provided references above. For the present study it is nevertheless important to know that the gridded simulations at $500 \times 500$ m resolution have not been specifically re-calibrated for the catchments investigated.
Instead, the spatially explicit version of PREVAH accesses a previously calibrated set of model parameters covering Switzerland which have been estimated using a regionalization approach (Viviroli et al., 2009c, a; Köplin et al., 2010). We also note that runoff rates from PREVAH are to be considered as the natural response of the grids within the catchments investigated, without any considerations of water diversions for hydropower, flood damping by (regulated) lakes or any kind of water use (Brunner et al., 2019d).

 **3 Methods**

## 3.1 Neural network architectures

We used two classes of temporal neural network models for runoff modeling, the long short-term memory model (LSTM, Hochreiter and Schmidhuber, 1997) and the temporal convolutional network (TCN, Bai et al., 2018). While the LSTM maintains an internal state that is updated dynamically, the TCN is based on a sparse and efficient 1D-convolution in the time domain. The latter is parallelizable in time and therefore computationally more efficient. We employ three approaches, described hereafter, to fuse the dynamic meteorological variables $\boldsymbol{x}_{t,c}$ at time $t$ and catchment $c$, with the static variables $\boldsymbol{s}_c$, independent of the temporal model used. The selection of the best fusion approach was part of the hyperparameter tuning (Sect. 3.3) and was performed independently of the model setup described in Sect. 3.5. Note that the following description of the model architectures is simplified and that the actual setup uses vectorized and efficient computation.

### 3.1.1 Prefusion with encoding

In the first approach, which we called *prefusion with encoding*, $\boldsymbol{x}_{t,c} \in \mathbb{R}^M$ (a vector of $M$ meteorological features) and $\boldsymbol{s}_c \in \mathbb{R}^S$ (a vector of $S$ static features) are, as part of the model training, encoded into $\boldsymbol{e}_{t,c} \in \mathbb{R}^D$, i.e., into a vector of length $D$ (the model dimensionality). The encoding is done using stacked feed-forward neural network layers $f_{\text{NN}_1}$ and $f_{\text{NN}_2}$, respectively. The two encodings are combined by element-wise addition, i.e., static encoding is added to each meteorological encoding equally, as shown in Eq. (1). The resulting combined encoding $\boldsymbol{e}_{t,c}$ is then fed into one or more temporal layers $f_{\text{TNN}}$ (Eq. (2)), yielding the temporal encoding $\boldsymbol{h}_{t,c} \in \mathbb{R}^D$, which is then decoded into a single value $q_{t,c}^* \in \mathbb{R}$ (Eq. (3)) by another stack of feed-forward neural networks $f_{\text{NN}_3}$.

$$\text{feature encode} \quad \boldsymbol{e}_{t,c} = f_{\text{NN}_1}(\boldsymbol{x}_{t,c}) + f_{\text{NN}_2}(\boldsymbol{s}_c) \tag{1}$$

$$\text{temporal encode} \quad \boldsymbol{h}_{t,c} = f_{\text{TNN}}(\boldsymbol{e}_{t,c}, \boldsymbol{e}_{t-1,c}, \ldots, \boldsymbol{e}_{t-k,c}) \tag{2}$$

$$\text{output decode} \quad q_{t,c}^* = f_{\text{NN}_3}(\boldsymbol{h}_{t,c}) \tag{3}$$

$$\text{output transform} \quad q_{t,c} = \log(1 + \exp(q_{t,c}^*)) \tag{4}$$

While the LSTM uses all input time steps, the TCN uses limited context $k$, depending on its hyperparameters. The decoded output is then transformed to the positive domain, $q_{t,c} \in \mathbb{R}_+$, using the softplus activation, as shown in Eq. (4). This output mapping is consistent across the fusion methods.

In this fusion approach, the potentially complex interactions of the dynamic and static input variables are injected prior to the temporal layer, presumably unloading some non-temporal interaction complexity from it. Note that the selection of model characteristics, such as number of hidden nodes and layers, was based on hyperparameter tuning (see next section).

### 3.1.2 Prefusion with repetition

In the second approach, *prefusion with repetition*, the static vector $\boldsymbol{s}_c$ is simply repeated in time and concatenated to the temporal input $\boldsymbol{x}_{t,c}$ (Eq. (5)). This combined encoding, based on a feed-forward neural network $f_{\mathrm{NN_4}}$, is then fed into the temporal module (Eq. (6)) and mapped to the output with another neural network $f_{\mathrm{NN_5}}$ as previously described and as shown in Eq. (7) and Eq. (8). This approach is conceptually similar to prefusion with encoding, but it leaves the learning of the non-linear interactions within the static inputs to the temporal layer.

$$\text{feature encode} \quad \boldsymbol{e}_{t,c} = f_{\mathrm{NN_4}}([\boldsymbol{x}_{t,c}, \boldsymbol{s}_c]) \tag{5}$$

$$\text{temporal encode} \quad \boldsymbol{h}_{t,c} = f_{\mathrm{TNN}}(\boldsymbol{e}_{t,c}, \boldsymbol{e}_{t-1,c}, \dots, \boldsymbol{e}_{t-k,c}) \tag{6}$$

$$\text{output decode} \quad q^*_{t,c} = f_{\mathrm{NN_5}}(\boldsymbol{h}_{t,c}) \tag{7}$$

$$\text{output transform} \quad q_{t,c} = \log(1 + \exp(q^*_{t,c})) \tag{8}$$

### 3.1.3 Postfusion with repetition

*Postfusion with repetition*, finally, first encodes the meteorological input $\boldsymbol{x}_{t,c}$ (Eq. (9)) with a feed-forward neural network $f_{\mathrm{NN_6}}$, runs the encoding through the temporal module (Eq. (10)). It then decodes the combined temporal encoding and static inputs $\boldsymbol{s}_c$ via repetition in time (Eq. (11)) by $f_{\mathrm{NN_7}}$, followed by the mapping to the positive domain (Eq. (12)). In this approach, the dynamics learned by the temporal layers cannot be modulated by the static variables.

$$\text{feature encode} \quad \boldsymbol{e}_{t,c} = f_{\mathrm{NN_6}}(\boldsymbol{x}_{t,c}) \tag{9}$$

$$\text{temporal encode} \quad \boldsymbol{h}_{t,c} = f_{\mathrm{TNN}}(\boldsymbol{e}_{t,c}, \boldsymbol{e}_{t-1,c}, \dots, \boldsymbol{e}_{t-k,c}) \tag{10}$$

$$\text{output decode} \quad q^*_{t,c} = f_{\mathrm{NN_7}}([\boldsymbol{h}_{t,c}, \boldsymbol{s}_c]) \tag{11}$$

$$\text{output transform} \quad q_{t,c} = \log(1 + \exp(q^*_{t,c})) \tag{12}$$

## 3.2 Model training and hyperparameter tuning

### 3.2.1 Data transformation

We transformed both the dynamic and static input features using $z$-transformation to have zero mean and unit variance. This process was executed individually for each cycle of cross-validation (see Sect. 3.4) and based on the specific training set assigned to that cycle. To maintain the target variable, i.e., runoff, within a positive range, its values were adjusted through normalization by dividing the values by the global 95th percentile derived from the training set.

### 3.2.2 Model optimization

The model parameters were updated using standard backpropagation (Amari, 1993) with the AdamW optimizer (Loshchilov and Hutter, 2019), a stochastic gradient descent method with adaptive first-order and second-order moments. As the objective

function, we used the mean squared error (MSE), defined as $\mathcal{L}_{\mathrm{MSE}} = \frac{1}{TC} \sum_{c=1}^{C} \sum_{t=1}^{T} (y_{t,c} - \hat{y}_{t,c})^2$, where $y_{t,c}$ is the normalized observation and $\hat{y}_{t,c}$ the respective predicted runoff at time $t$ of $T$ number of time steps and catchment $c$ among $C$ catchments. Optionally, we considered the square root-transformed prediction and target to reduce the right-skewness of the distribution and therefore to facilitate the training.

The training sets were constructed with the goal of ensuring equal representation of each catchment, regardless of the number of observations available from each. To do this, we iteratively selected samples from each catchment in a randomized order. We refer to one complete iteration through all catchments as an "epoch". For each catchment, a two-year period was randomly selected, ensuring at least 30 days of runoff data were present. Additionally, a one-year lead-in phase was introduced for model spin-up, which was not factored into the optimization calculations.

Throughout the model training phase, we used minibatches of 32 samples. A minibatch is a subset of the training data used in each step of the gradient descent process to update the model's parameters. This approach strikes a balance between computational efficiency and the stochastic nature of the training, allowing more frequent updates and efficient use of parallel processing. For validation and testing, the complete time series was processed in each evaluation epoch, optimizing for efficiency since no parameter updates were needed in these phases.

## 3.3  Hyperparameter tuning

Hyperparameter tuning, an essential step in enhancing a deep learning model's performance, was conducted systematically. This involves identifying the best combination of preset parameters, like the learning rate or the number of neurons per layer, to optimize model performance. We refer to Appendix A1 for a comprehensive list of hyperparameters used. We used the initial cycle of our cross-validation (see Sect. 3.4) process for this tuning. The hyperparameters were tuned using the Optuna framework (Akiba et al., 2019). After the evaluation of 15 random hyperparameter combinations, 45 further configurations were suggested iteratively using a Bayesian surrogate model based on the tree-structured Parzen estimator (TPE) algorithm (Bergstra et al., 2011). As some configurations may perform poorly in the early training phase, we used hyperband pruning to stop such unpromising runs early on without wasting resources (Li et al., 2018). With the optimal hyperparameters determined, we completed the full cross-validation process described in the next section.

## 3.4  Cross-validation

We carefully designed a $k$-fold cross-validation setup for a fair model evaluation and to assert the high quality of the final reconstruction product. The 98 training catchments were randomly divided into $k = 8$ sets and iterated over such that each set was used as both validation and testing once during the cross-validation process (Fig. 2a). While the training data are used for optimizing the model and validation data are used to monitor model generalization during training, the test data serve for the final model evaluation. The remaining 169 catchments, which are more impacted by anthropogenic factors, were used optionally as additional training catchments – but never to evaluate the model. In addition, the time domain was split into training, validation, and test periods (Fig. 2b). These periods were kept fixed during cross-validation. The temporal splitting was chosen to be representative of the model's temporal interpolation and extrapolation skills. At the same time, the validation and

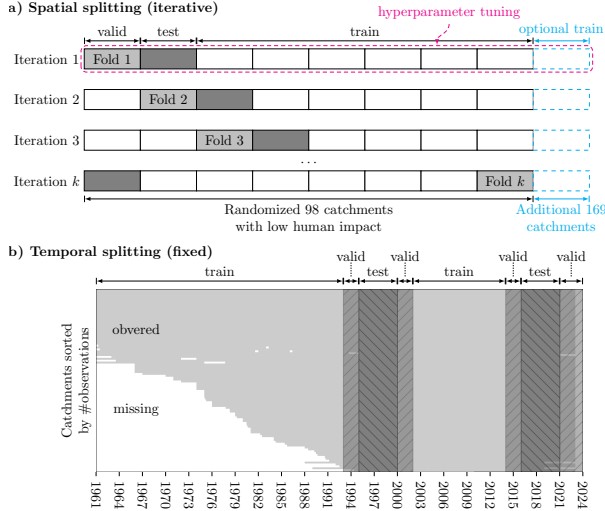

**Figure 2.** Cross-validation scheme: Panel a) shows the spatial domain, consisting of 98 catchments in Switzerland, which we randomly divided into $k = 8$ sets. These sets were used iteratively, allowing each to serve as a validation and test set once. Catchments significantly affected by human actions were included in the training phase optionally to enrich model learning, but they were consistently excluded from the validation and test phases to maintain the focus on natural runoff patterns. In Panel b), the time domain is delineated into fixed training, validation, and test periods. The used sets are the intersections of these spatial and temporal splits. With the initial iteration, hyperparameter (HP) tuning was performed, and the best HPs identified were then applied in subsequent cross-validation steps. By evaluating the model on the test sets, we comprehensively assessed the model's skill in generalizing across both spatial and temporal dimensions.

test periods should contain minimal missing data, to not give more emphasis to catchments with more observations. Therefore,

we selected two five-year blocks of test data, one from (the beginning of) 1995 to (the end of) 1999, and one from 2016 to 2020. The test ranges were separated from the training set to ensure minimal data leakage. This relates to the fact that autocorrelation in time series data can lead to overfitting because it causes models to mistake random patterns in the data as significant. In addition, the buffer added after every temporal block avoids overlap of the test set spin-up period of one year with the training set, which would again encourage overfitting. Note that the validation set was not separated by a buffer from the training set to

avoid discarding any observations and because the final evaluation was done on the test set. Overall, the spatial and temporal data splitting was a trade-off between computational efficiency, autocorrelation concerns, and data limitations.

In each of the $k$ iterations, six catchment sets were used for training, i.e., for optimizing the neural network parameters, while one set was used for validation and one was held out for testing. After an epoch, i.e., one full iteration through the training data, the loss was computed on the validation set. The loss was monitored and training was interrupted if the loss on

the validation set was increasing over a given number of epochs (the "patience"). The best model in terms of the validation loss was then restored and used for prediction on the test set. This routine is called "early stopping" and reduces overfitting (Yao

et al., 2007). The final predictions on the test set were then used for model evaluation. As each catchment set was the test set once, we obtained independent test predictions for each of the 98 catchments.

## 3.5 Factorial experiment and model evaluation

In this section, we describe the model setups tested in a factorial experiment and the model evaluation procedures. We selected the best-performing model based on the median Nash-Sutcliffe modeling efficiency (NSE, Nash and Sutcliffe, 1970) across catchments, evaluated on the test set. The NSE as defined in Eq. (13) is calculated on catchment level:

$$\text{NSE} = 1 - \frac{\sum_{t=1}^{T}(y_t - \hat{y}_t)^2}{\sum_{t=1}^{T}(y_t - \overline{y})^2} \quad , \tag{13}$$

where $y_t$ is the observed and $\hat{y}_t$ the simulated runoff at time $t$ of $T$ total time steps, and $\overline{y}$ is the mean of the observed time-
series. The NSE can take values from $-\infty$ to 1, where values above 0 indicate that the predictions are better than taking the mean of the observations, and 1 means perfect prediction. Note that the NSE is closely related to the R-squared, but the NSE normalizes the sum of the squared residuals by the catchment variance instead of the global variance. Hence, the NSE does not give more emphasis to catchments with larger variance and is, therefore, also sensitive to catchments with low dynamic range.

Different model setups were tested in a factorial experiment, and each combination of the factors was evaluated. A first factor
determines the temporal component of the overarching model architecture: $\{\text{LSTM}, \text{TCN}\}$. A next factor determines whether the target variable and predictions are transformed using the square root, in order to reduce the skewness of the distribution: $\{\text{T}_{\text{none}}, \text{T}_{\text{sqrt}}\}$. We tested the inclusion of the 169 optional catchments (267 in total with the 98 default catchments) in the training set, compared to the 98 only (Fig. 2a): $\{\text{C}_{98}, \text{C}_{267}\}$. The last factor concerns the usage of static input variables. Due to the relatively large number of catchment properties (28), we alternatively used dimensionality-reduced static features.
Using principal component analysis (PCA, Wold et al., 1987), all static features except catchment area were compressed into 5 components, which represent 66 % of the variance. Catchment area was always treated as a separate static input, as we consider it to be a key input feature. Hence, we either use catchment area only, a dimensionality-reduced version of the static variables using PCA, or all static variables described in the data section: $\{\text{S}_{\text{area}}, \text{S}_{\text{PCA}}, \text{S}_{\text{all}}\}$.

This yields a total of 24 models, and for each of them, independent hyperparameter tuning and cross-validation were per-
formed. Note that the fusion strategy for static and dynamic features, introduced in Sect. 3.1, was not considered a factor here, but was part of the hyperparameter tuning.

To better understand the error structure, we also evaluate the MSE decomposition into bias, variance, and phase error (Kobayashi and Salam, 2000; Gupta et al., 2009):

$$e_{\text{MSE}} = \overbrace{(\mu_{\hat{y}} - \mu_y)^2}^{e_{\text{bias}}} + \overbrace{(\sigma_{\hat{y}} - \sigma_y)^2}^{e_{\text{variance}}} + \overbrace{2\sigma_{\hat{y}}\sigma_y(1 - r)}^{e_{\text{phase}}} \quad , \tag{14}$$

where $\mu$ is the mean and $\sigma$ is the standard deviation of the simulations $\hat{y}$ and the observations $y$, and $r$ is the linear correlation coefficient between them. The squared bias $e_{\text{bias}}$ reflects the model fit in terms of the average and the variance error $e_{\text{variance}}$ in terms of the scale. The phase error $e_{\text{phase}}$ measures the reproduction of timing, i.e., how well the dynamics are matched regardless of bias and scale.

### 3.6 Runoff reconstruction

To achieve a complete contiguous reconstruction from 1962 to 2023 for the small to medium-sized catchments with national coverage (Fig. 1), we used the best-performing model from the cross-validation. The best model was selected based on median test set NSE across catchments. From the ensemble members from the eight-fold cross-validation, we obtained eight reconstructions with full coverage, of which we use the median (average of the two middle values) as the final data product. The year 1961 was removed from the reconstruction as it served as spin-up.

## 4 Results

### 4.1 Catchment level performance and benchmarking

In this section, we evaluate the model performance at catchment level and compare the data-driven models. All analysis is, unless stated otherwise, based on the test set (Fig. 2), i.e., spatially and temporally distinct data. Due to the iteration over the catchment groups in the cross-validation, each catchment was in the test set once. The fixed splitting of the time domain, however, restricts our evaluation to the test periods, i.e., January 1995 to December 1999, and January 2016 to December 2020. Throughout the evaluation, we use the hydrological model PREVAH as a benchmark.

#### 4.1.1 Model performance

To understand the capabilities of our model to represent daily runoff at catchment level, we evaluate the model performance first. Figure 3 presents the empirical cumulative density functions for different metrics across the 98 catchments. Models based on the TCN architecture are depicted in blue, those using LSTM networks in red, and the PREVAH model is represented in black. The model with the best performance is emphasized using a thicker line. Panel a focuses on the Nash-Sutcliffe Efficiency (NSE), while panels b–d provide a detailed breakdown of the Mean Squared Error (MSE) into its components – squared bias, variance error, and phase error – as previously introduced.

Overall, we observed a large variance in performance across model setups in terms of catchment-level NSE, and the TCN-based models performed worse than the LSTM in general. This is mainly due to the two best-performing LSTM models (also see model-wise NSE in Appendix A, Tab. A2 & A3). The best-performing LSTM ($\text{LSTM}_{\text{best}}$) achieved a median NSE of 0.76. The MSE decomposition shown in Fig. 3b–d indicates that the $\text{LSTM}_{\text{best}}$ (thick red line) is among the best models in terms of all error components. The phase error contributed most to the overall error by a wide margin, signifying that representing the timing of the runoff is more challenging than representing the average and the scale.

The best-performing setup was $\{\text{LSTM}, \text{S}_{\text{all}}, \text{C}_{267}, \text{T}_{\text{sqrt}}\}$, i.e., with all static features, additional training catchments, and with square root transform of the target, paired with the LSTM architecture. This model performed only marginally better than $\{\text{LSTM}, \text{S}_{\text{all}}, \text{C}_{98}, \text{T}_{\text{sqrt}}\}$, i.e., the one not using the additional catchments for training. These models both worked best with the prefusion with encoding approach (see Sect. 3.1). An overview of all model setups and their performance is provided in Appendix A1, and a short discussion of the factorial experiment can be found in Appendix A2.

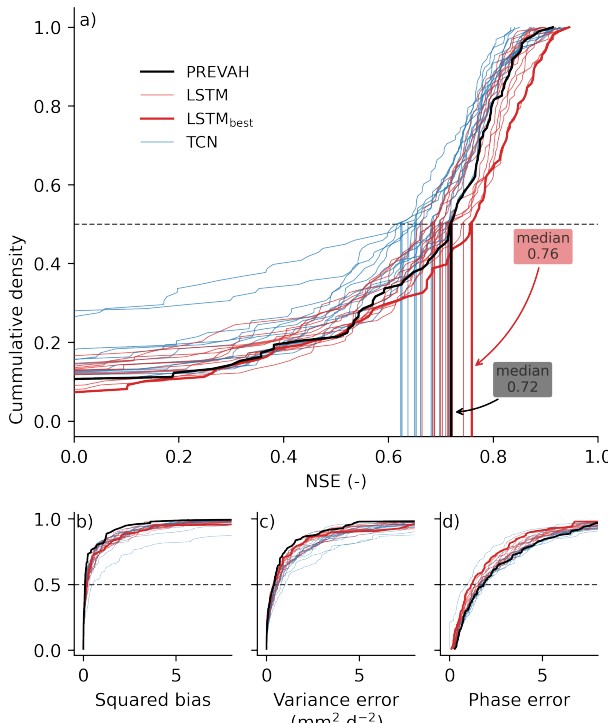

**Figure 3.** The catchment-level model performance across the 98 catchments evaluated on the test set that was not used for model calibration. We show different versions of the data-driven models corresponding to model setups, blue represents the convolution-based architectures (TCN), and red the LSTM architectures. The PREVAH model (black solid line) serves as a benchmark. The best-performing model (LSTM$_{\text{best}}$), used for the reconstruction, is highlighted. The y-axis represents the cumulative probability density, i.e., the fraction of catchments that have given value or lower. Panel a) shows the Nash-Sutcliffe modeling efficiency (NSE) with median values as vertical lines. The panels b)–d) show squared bias, variance error, and phase error, respectively. Note that here, other than for NSE in panel a), lower values are better. The x-axes are truncated.

The PREVAH model achieved a median NSE of 0.72 (Fig. 3a), which is marginally lower than LSTM$_{\text{best}}$ (NSE of 0.76), yet better than some of the other data-driven models. The PREVAH model showed similar bias and variance error (Fig. 3b–d) to those of LSTM$_{\text{best}}$ in terms of the median, yet it seems to be more robust in representing these aspects, as the LSTM$_{\text{best}}$ lags behind in the larger errors. Regarding the phase error, in contrast, the data-driven models in general and the LSTM$_{\text{best}}$ in particular clearly outperformed PREVAH across catchments.

Next, we investigate the spatial distribution of the errors. First, we notice that the performance of LSTM$_{\text{best}}$ in terms of NSE, shown in the top-left panel Fig. 4, does not exhibit a clear spatial pattern. Yet, the model seems to struggle with some particular catchments. Interestingly, these are the very catchments where PREVAH outperformed LSTM$_{\text{best}}$ clearly (compare Fig. 4 top-left panel dark blue values to lower-left panel dark red values).

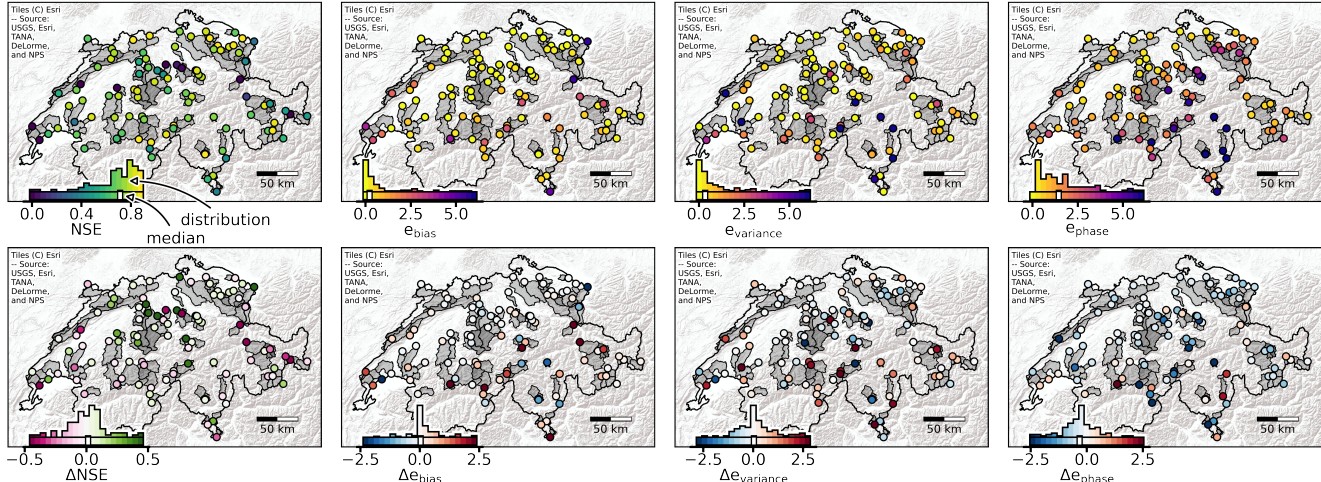

**Figure 4.** Spatial catchment-level performance of our best-performing model ("LSTM$_{\mathrm{best}}$") contrasted to the PREVAH hydrological model. The top row shows the performance of LSTM$_{\mathrm{best}}$, with the Nash-Sutcliffe modeling efficiency (NSE) in the most-left panel, and bias ($e_{\mathrm{bias}}$), variance ($e_{\mathrm{variance}}$), and phase ($e_{\mathrm{phase}}$) error, in the remaining panels. Note that in the top panel, yellowish colors indicate better performance, i.e., for NSE, a larger number is better and for the error components, lower numbers are preferred. The bottom row shows the performance difference of LSTM$_{\mathrm{best}}$ minus PREVAH. Here, reddish colors indicate that PREVAH performs better than LSTM$_{\mathrm{best}}$, which is, for NSE, negative values, and for the error components, positive values. The inset histograms represent the distribution of catchment metrics, and the white bar indicates the median of the distribution, per panel. The evaluation is performed on the test set, but all catchments are in this set once in our cross-validation setup.

To understand how these spatial patterns are linked to catchment properties, we performed an exploratory analysis. First, we identified the tails of the distributions (inset histograms in Fig. 4, and PREVAH performance, not shown) using the 10th and 90th percentiles. We then compared properties of catchments in the tails to the "normal" group (between the 10th and 90th percentiles) using the two-sided, non-parametric Mann-Whitney U test with a significance level of $\alpha = 0.1$ (Mann and Whitney, 1947). The analysis was restricted to a subset of catchment properties: mean and variance of runoff, elevation, catchment area, and water body fraction. Here, we report the most notable findings from this ad-hoc analysis.

For LSTM$_{\mathrm{best}}$, poor performance (NSE below 0.14) was observed in catchments with low mean and variance of runoff, whereas good performance (NSE above 0.86) was achieved in catchments with high runoff variance. Similarly, PREVAH struggled (NSE below 0.08) in catchments with low runoff mean and variance, as well as in low-elevation, lake-dominated conditions, but performed well (NSE above 0.84) in catchments with high runoff mean, large catchment areas, and minimal lake presence. As expected, the bias of LSTM$_{\mathrm{best}}$ was low in catchments with low runoff mean and variance, with variance error increasing in high runoff variance conditions. Phase error for LSTM$_{\mathrm{best}}$ was lowest in catchments with low runoff mean and variance and large catchment area.

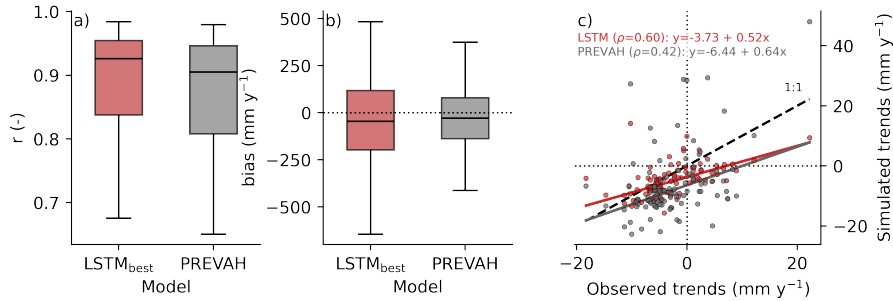

**Figure 5.** Catchment-level evaluation at the annual scale. a) The Pearson correlation (r) and b) bias in $\mathrm{mm\,y^{-1}}$ distribution across 98 training catchments evaluated on the test set. c) The simulated annual runoff trends compared to observations. The points represent the linear trend (found by robustified least squares fit) of single catchments. Note that for the trend calculation, the time range from 1995 (start of first test period) to 2020 (end of second test period) was used. The inset equation shows the linear least square fit and the corresponding rank correlations.

Significant differences in NSE performance were observed between the two models in catchments with low runoff variance PREVAH outperformed $\mathrm{LSTM_{best}}$ (NSE improvement above 0.28) in catchments with both low runoff mean and variance. Conversely, $\mathrm{LSTM_{best}}$ clearly outperformed PREVAH (NSE improvement above 0.28) in low-elevation, lake-dominated catchments that also had low runoff variance.

### 4.1.2 Annual variability and trends

For a reconstruction product, it is crucial to adequately represent yearly variability and long-term trends. We, therefore, evaluate this aspect on annual runoff aggregates (Fig. 5). The best-performing model, $\mathrm{LSTM_{best}}$, represented the interannual variability (Fig. 5a), quantified as the Pearson correlation coefficient between the annual values for each catchment, well with a median of $r = 0.93$, and with 75 % of catchments above $r = 0.85$. The bias averages close to zero and for 50 % of the catchments, it was in the range of -250 to 250 $\mathrm{mm\,y^{-1}}$ (Fig. 5b). On the interannual variability, PREVAH showed a slightly lower correlation (Fig. 5a) across catchments with a median of $r = 0.91$. In terms of bias, PREVAH performed marginally better with a median closer to zero and a lower spread (Fig. 5b).

Figure 5c illustrates how the models captured spatial patterns of annual trends between January 1995 and December 2020 (Fig. 5c). The agreement was calculated by first computing the catchment-level linear trends for the observations and the simulations by PREVAH and $\mathrm{LSTM_{best}}$ independently using the robust Theil-Sen estimator (Sen, 1968). Then, we fit a regression between the observed and estimated trend slopes by the two models using robust regression with Huber weighting and the default tuning constant of $c = 1.345$ (Huber and Ronchetti, 2009). This approach reduces the impact of outliers by giving lower weight to large residuals. For quantifying the alignment of the simulated trends, we use Spearman correlation ($\rho$), which is relatively robust against outliers. While the $\mathrm{LSTM_{best}}$ represented the spatial patterns of the linear trend relatively well with a correlation of $\rho = 0.60$, PREVAH achieved a correlation of $\rho = 0.42$. Both models underestimated the strength of negative

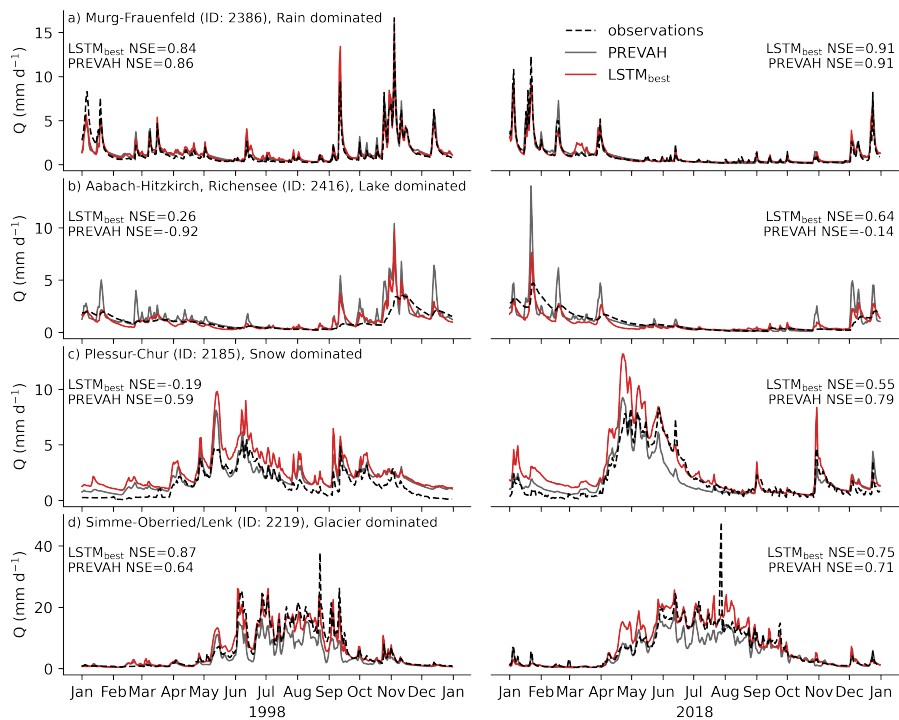

**Figure 6.** Daily runoff Q in $\mathrm{mm\,d^{-1}}$ for four selected catchments and two distinct years selected from the test periods. Observed (dashed black line), simulated by PREVAH (grey), and out-of-catchment prediction of the data-driven $\mathrm{LSTM_{best}}$ (red) are shown. The catchments were selected to represent different modeling challenges: a) rainfall, b) lake, c) snow, and d) glacier dominated. The inset NSE values represent the model performance for the selected year.

and positive trends with slopes of 0.52 ($\mathrm{LSTM_{best}}$) and 0.64 (PREVAH), and they exhibited small negative biases of -3.73 ($\mathrm{LSTM_{best}}$) and -6.44 (PREVAH) $\mathrm{mm\,y^{-1}}$.

### 4.2 Qualitative evaluation of selected catchments

To understand how the models represent different hydrological regimes, we perform a qualitative comparison for a selection of catchments. We selected single catchments that are either dominated by a) rainfall, b) lakes, c) snow, or d) glaciers for this purpose (Fig. 6). These example catchments serve as means for qualitative model comparison, and we do not expect these insights to directly generalize across catchments.

The rainfall-dominated catchment, the Murg at Frauenfeld (ID: 2386), is located in the Northwest of Switzerland on the
Swiss Plateau, with an area of $\sim$200 $\mathrm{km^2}$ and an average elevation of $\sim$600 m. Maximum snow water equivalent (SWE) has been below 80 mm in the past years (Höge et al., 2023) and was not considered to affect runoff for most days of the year. The lake-dominated catchment, the Aabach at Hitzkirch (ID: 2416), is located in the central Pre-Alps, with an area of $\sim$70 $\mathrm{km^2}$ and an average elevation of $\sim$600 m. The gauging station is located just a few hundred meters after the outflow of the $\sim$5 $\mathrm{km^2}$

Lake Baldegg, which damps runoff peaks and also affects the low flow regime. The snow-dominated catchment, the Plessur at Chur (ID: 2185), is located in the Eastern Swiss Alps, with an area of ∼250 km² and an average elevation of ∼1900 m (from ∼500 m to ∼3000 m). Maximum SWE varied in the past 25 years between 200 and 500 mm. There are no large glaciers in this area that could influence runoff. The glacier-dominated catchment, the Simme at Oberried/Lenk (ID: 2219), is located in the Western Swiss Alps, with an area of  35 km² and an average elevation of ∼2300 m (from ∼1000 m to ∼3200 m). About 25 % of the area is covered by glaciers. Maximum SWE varied in the past 25 years between 400 and 1100 mm.

For the rainfall-dominated catchment, PREVAH and $\text{LSTM}_{\text{best}}$ showed similar behavior (Fig. 6a) and both models could reproduce the runoff peaks and overall patterns. For the lake-dominated catchment, the $\text{LSTM}_{\text{best}}$ outperformed the PREVAH model in terms of NSE (Fig. 6b). Visual inspection shows high peaks in PREVAH simulations, which indicate missing buffering dynamics in lakes. This is not surprising, as PREVAH does not represent lakes explicitly, while the LSTM can learn the buffering implicitly via the catchment properties, among which the fraction of water bodies may be the most relevant one. For the snow-dominated catchment shown in (Fig. 6c), the PREVAH model managed to represent the runoff processes better in 1998, and similar in the year of 2018. Here the $\text{LSTM}_{\text{best}}$ overestimates runoff in general, and particularly peaks in the summer. Snowmelt responds strongly to radiation, which was not included as a driver of the LSTM. Further, snow-related processes are spatially heterogeneous, depending on elevation and aspect. The lumped LSTM model cannot resolve these processes at subcatchment level, while the PREVAH model operates on a high-resolution grid. Although worse in terms of NSE, the LSTM managed to better represent the snowmelt in 2018, possibly because snow was already melted away in the PREVAH simulation. In a glacier-dominated catchment, finally, $\text{LSTM}_{\text{best}}$ represented the runoff patterns slightly better than PREVAH.

## 4.3 Runoff reconstruction

The reconstruction of daily runoff from 1962 to 2023, referred to as CH-RUN, was conducted with the best-performing model based on prior analysis. The final estimate was calculated as the median across the eight ensemble members from the cross-validation. Figure 7 shows the annual runoff as quantiles relative to the reference period from 1971 to 2000. The quantiles were calculated per catchment comparing the annual values to the empirical distribution of the reference period. Turquoise colors indicate that, for a given catchment, the yearly average runoff is rather large compared to the reference, and brown colors signify dry years.

The reconstruction suggests that dry years with similar intensity compared to the conditions of the 21st century were already present in the 1960s and 70s (e.g., in 1964 and 1976). However, the frequency of dry years increased significantly – and that of wet years substantially decreased – according to the model estimates.

Figure 8 shows the annual national runoff anomalies and corresponding trends for CH-RUN. According to the CH-RUN reconstruction, the recent dry conditions are matched by values in the 1960s and 70s in terms of amplitude, while the frequency of dry years increased, and that of wet years decreased. Extremely dry years (exceeding the 0.1 quantile of the reference period) were absent in the 1980s and 1990s, while wet years (exceeding the 0.9 quantile of the reference period) were more frequent during this period. The last extremely wet year occurred in 1999, and the driest year was 2022.

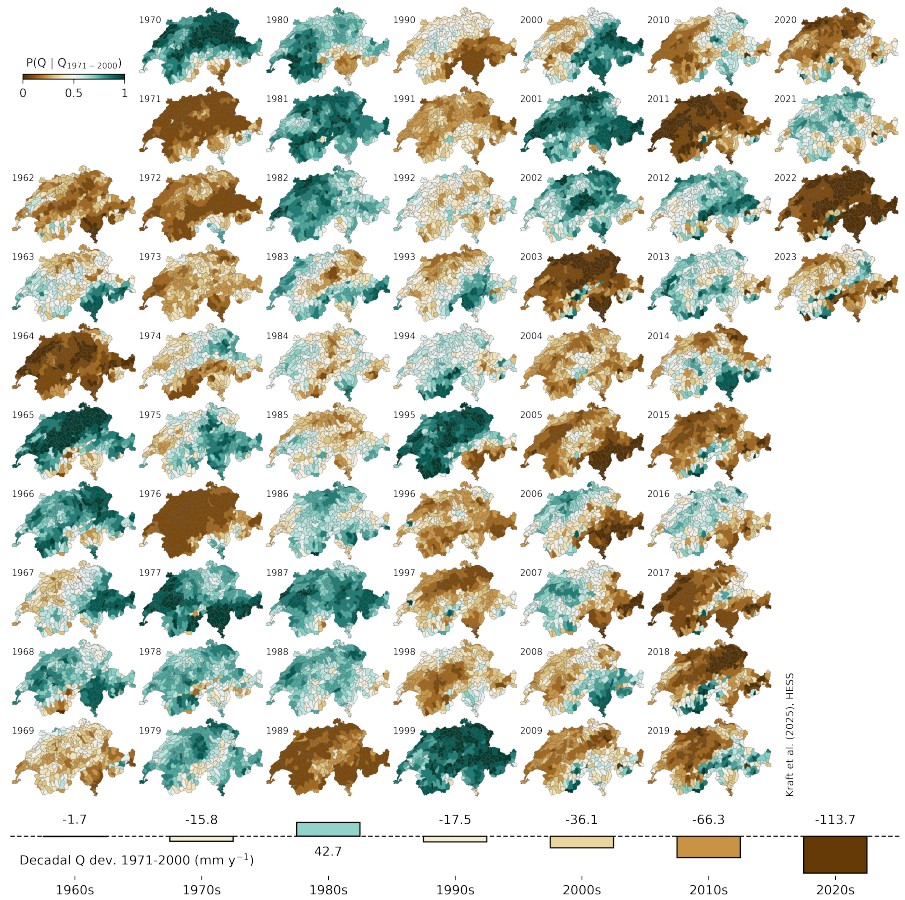

**Figure 7.** Spatially contiguous reconstruction of runoff from 1962 to 2023 from the CH-RUN reconstruction. The maps represent the yearly catchment-level runoff quantiles relative to the reference period (1971 to 2000) empirical distribution. The bottom bars show the decadal deviation in $\mathrm{mm\,y^{-1}}$ of the national-level runoff relative to the reference period (1971 to 2000).

In Fig. 9, the decadal mean values are disaggregated into seasonal patterns. Here, average annual sums across decades are shown, again relative to the reference period from 1971 to 2000. The decadal means again point at strong trends towards less runoff on a yearly scale. In the winter months December to February (DJF), we see a slight tendency towards less runoff north of the Alps, while the 2020s exhibit more runoff in the Pre-Alps. From March to May (MAM), northern Switzerland, the Pre-Alps, and the canton of Ticino show a clear trend towards drier conditions. Even more pronounced, June to August (JJA) reveal a tendency towards less runoff in the central Alps with higher altitudes and the canton of Ticino. From September to November (SON), the patterns are again less distinct, yet there is a general trend towards drier conditions in most catchments.

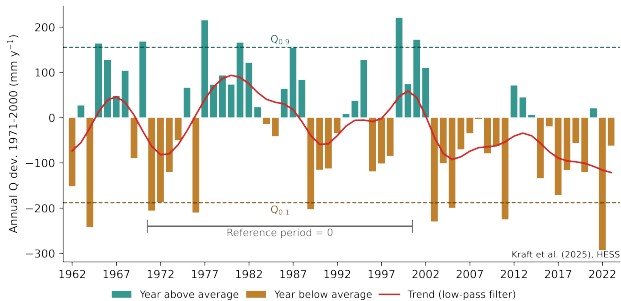

**Figure 8.** Annual runoff anomalies for Switzerland from 1962 to 2023 from the CH-RUN reconstruction. The bars represent the CH-RUN annual deviation in $\mathrm{mm\,y^{-1}}$ of the national-level runoff relative to the reference period (1971 to 2000). The aggregation from catchment to national level was done with area-weighted mean. Positive anomalies are depicted in turquoise, negative ones in brown. The solid red line represents the trend, i.e., the low-pass filtered signal using a Gaussian filter with a standard deviation of $\sigma = 2$. The dashed lines are the 0.1 and 0.9 quantile of the reference period, respectively.

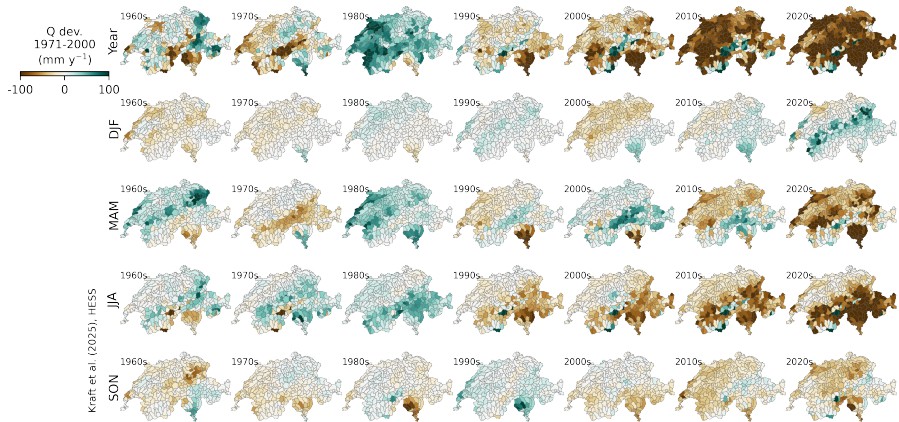

**Figure 9.** Decadal evolution of spatially contiguous reconstruction of runoff from 1962 to 2022 by season from the CH-RUN reconstruction. The maps represent the decadal average catchment-level runoff in $\mathrm{mm\,y^{-1}}$ relative to the reference period (1971 to 2000). From top row to bottom: Year: full year; DJF: December to February; MAM: March to May; JJA: June to August; SON: September to November.

## 5 Discussion

### 5.1 Neural network architectures and the role of data

The better performance of the LSTM compared to the TCN (Fig. 3) is somewhat surprising, as the latter has been reported to perform well in time series prediction settings (e.g., Zhao et al., 2019; Catling and Wolff, 2020; Yan et al., 2020). The difference in performance can be traced back to the two best-performing LSTMs (Fig. 3, Tab. A2 & A3) and hint at better capabilities of the LSTM to represent interactions between meteorological and static features, under these data-limited conditions. It seems

that the LSTM is more data-efficient than the TCN, which is also supported by the lower number of tunable parameters used by the former (see Tab. A2 & A3). It might, therefore, be possible that the TCN can compete with the LSTM architecture if more training data are available. Other deep learning approaches for modeling time series exist, among which transformer-based architectures (Vaswani et al., 2017) have become popular recently (Lim et al., 2021; Zhou et al., 2021; Xu et al., 2023). Due to the powerful and complex encoder-decoder structure, these models unfold their potential especially in forecasting settings and

with large amounts of training data. Given the relatively low amounts of training data available for the study domain, we do not expect significant improvement by using such architectures. Nevertheless, exploration of this architecture may hold potential for improved reconstruction in the future.

    In runoff modeling, integrating catchment properties with meteorological features is a prevalent approach (e.g., Kratzert et al., 2019). We found that the most effective data fusion method involved channeling static variables through the LSTM's

temporal layers while handling some non-temporal interactions in an upstream encoding layer. Our LSTM successfully learned the complex interactions between static and meteorological features, extending its applicability to untrained catchments and time ranges. Enhancing the model's predictions was achieved by incorporating a broader range of catchments and fully utilizing catchment properties (Fig. A1), acknowledging that a wider input feature space necessitates more data. This finding aligns with our previous diagnosis of spatial information limitations due to the relatively small number of training catchments. Although

the importance of data has been well reported (Kratzert et al., 2018; Gauch et al., 2021b), these results reaffirm the value of additional data in enhancing model performance. Consequently, we recommend exploring methods to incorporate more training data, such as transfer learning from other tasks (Sadler et al., 2022) or other regions (Pan and Yang, 2010; Yao et al., 2023; Xu et al., 2023), for example from large-scale datasets (Kratzert et al., 2023; Do Nascimento et al., 2024), and considering alternative data sources like bottom-up data mobilization efforts (Do et al., 2018; Gudmundsson et al., 2018; Nardi et al., 2022;

Kebede Mengistie et al., 2024) as promising avenues for future research.

    It is encouraging to see that the LSTM did manage to implicitly learn complex runoff dynamics across hydrological regimes (Fig. 6). The data-driven model has learned buffering effects by lakes and, to a certain extent, runoff-generating processes related to snow and, possibly, also glaciers. Similar behavior has been reported before. Kratzert et al. (2019) and Lees et al. (2022), for example, reported that an LSTM was able to represent long-term snow dynamics. We expect potential for improve-

ment by better representing buffering processes via routing of the runoff (Bindas et al., 2024) and by an improved representation of snow and glacier processes. This can be achieved via the combination of physically-based and data-driven modeling (Reichstein et al., 2019), e.g., by directly employing physical constraints in an end-to-end hybrid physics-machine learning setup (Kraft et al., 2022, 2020; Höge et al., 2022), by penalizing physically implausible simulations during training (Daw et al., 2021), or by regularizing the model with auxiliary tasks (Sadler et al., 2022), to only name a few.

## 5.2  Comparison with the PREVAH model

Although some neural networks outperform the PREVAH model (Fig. 3), the differences in terms of NSE were small. The marginally better representation of runoff mean and amplitude by PREVAH makes sense intuitively, as the data-driven model has a very limited number of training catchments to learn spatial features from. Equivalently, the better representation of

temporal patterns by the LSTM could be explained by the fact that it has access to long time-series to learn the dynamics from. It is not surprising that machine learning can outperform physically-based models in runoff prediction, as this has been demonstrated in previous studies (e.g., Kratzert et al., 2018; Lees et al., 2021; Gudmundsson and Seneviratne, 2015; Ghiggi et al., 2021). However, in this study, we used a limited amount of meteorological drivers compared to the needs of the PREVAH model. Furthermore, the PREVAH model is an expert model that is using carefully regionalized parameters for the study domain (Viviroli et al., 2009c). As PREVAH is providing the natural discharge within the catchment domain, it is not able to capture the dampening effect provided by lakes (Fig. 6b). The LSTM is able to cope with such effects as part of its global calibration result.

From the analysis of the spatial patterns of the model performance (Fig. 4), we learned that the LSTM$_{best}$ encountered challenges with dry catchments that have both low runoff mean and variance. This was not surprising due to the high signal-to-noise ratio in runoff observations and the sensitivity to minor variability in the meteorological variables and catchment properties in dry catchments. Similarly, PREVAH struggled with dry conditions, but still it performed clearly better under such conditions. In contrast, LSTM$_{best}$ represents lake-dominated catchments with low elevation significantly better. This was expected, as PREVAH does not represent lake processes, and therefore, it cannot properly represent their dampening effect. The interaction with elevation could be explained by the fact that the largest lakes in Switzerland are at medium-to-low elevation.

The PREVAH model is already used successfully for reconstruction (Otero et al., 2023) and future climate scenarios (Laghari et al., 2018; Brunner et al., 2019b). With the objective of real-time monitoring, long-term reconstruction, and potentially an efficient simulation of climate scenarios in mind, we consider the similar performance compared to the benchmark sufficient. The similar capability of representing interannual patterns by the LSTM$_{best}$ and the slightly better fidelity of trends (Fig. 5) is, especially given the lowered data requirements, encouraging. We want, however, to state here upfront that a process-based hydrological model has advantages over a data-driven model, such as interpretability and physical consistency.

## 5.3 Plausibility of the runoff reconstruction product

The reconstruction of runoff back to the early 1960s for Switzerland is a novelty enabled by the reduced data needs of our deep learning-based approach compared to the PREVAH model. Here, we evaluate the plausibility of the simulated patterns based on Fig. 7-9 by contrasting them to prior knowledge.

The overall trend towards drier conditions simulated by our data-driven model aligns with independent studies. This has been reported widely for Europe (Orth and Destouni, 2018; Hanel et al., 2018) and Switzerland specifically (Hohmann et al., 2018; Brunner et al., 2019b; Henne et al., 2018). Interestingly, the results reveal that the runoff anomalies of the 2022 drought (e.g., Toreti et al., 2022; Schumacher et al., 2024) were larger than that of the well-documented 2003 drought (e.g., Ciais et al., 2005; Rebetez et al., 2006; Seneviratne et al., 2012). The identified drying trend in the summer season is consistent with a reported increase in agro-ecological droughts in West-Central Europe in the latest report of the Intergovernmental Panel on Climate Change (Arias et al., 2023), and may indicate the presence of a drying trend in streamflow in this region, which was assigned low confidence at the time of the IPCC report (Seneviratne et al., 2021).

In the winter months, an increase in runoff in the Pre-Alps may be linked to an earlier onset of snowmelt (Vorkauf et al., 2021). In the same region and other medium-altitude areas such as the Jura sub-alpine mountain range, runoff is decreasing in spring. This could be related to a combination of a trend towards lower snowmelt due to less snowfall during winter (Matiu et al., 2021) and an earlier onset of snowmelt due to warmer temperatures mentioned previously. The Alps are, supposedly, similarly affected by those effects, yet the onset of thawing is delayed due to higher altitudes, and hence we see the main contribution to negative trends in later summer. In Ticino, a strong trend towards warmer temperatures has been reported, although precipitation seems to not show significant trends (Reinhard et al., 2005). The negative trend in summer is likely caused by both a lack of snowmelt and an increase in evapotranspiration via warmer air temperatures, which can have a significant impact on runoff (Teuling et al., 2013; Goulden and Bales, 2014).

## 5.4 Potential applications

Other than catchment-level observations, the spatially and temporally complete reconstruction provides a tool for studying runoff beyond the observational horizon and also for ungauged catchments. The focus on catchments with low human impacts during model training allows the investigation of physical processes in isolation. This is an advantage for climate-focused studies, as it is challenging and often not possible to disentangle effects from human water use from physical effects associated with human-induced climate change. We encourage researchers to use the CH-RUN product for trend analysis and to understand the drivers of the simulated patterns. We further see potential in using CH-RUN as an independent benchmark dataset for hydrological models: It is challenging to understand the different sources of uncertainty during model development. Having a methodologically independent benchmark dataset can help disentangle methodological from data limitations.

With our data-driven approach, we achieve a speedup by a factor of 600 compared to PREVAH, assuming PREVAH is run parallelized across 100 central processing units (CPUs) and the CH-RUN is employed on a high-performance graphics processing unit (GPU). The reconstruction for the entire domain took approximately 20 seconds for one ensemble member on a NVIDIA A100 GPU. This speedup enables computationally cheap real-time monitoring of runoff on a national scale. In addition, the model can be fed with meteorological forecasts, which would enable early warning of floods and droughts. A common use case for hydrological models is running scenarios, i.e., to simulate responses to a changing climate or to attribute runoff patterns to anthropogenic forcing. However, running scenarios with physically-based models is computationally expensive, which limits ensemble size and forecast horizon. The speedup compared to a traditional hydrological model allows running thousands of scenarios with ease. The application in early warning and running scenarios must be carefully examined and may require further calibration steps, but holds potential for understanding and mitigating climate change impacts in the near future.

## 5.5 Limitations

In the evaluation at the catchment level, it was observed that the CH-RUN model, although effective in general, faces challenges under certain conditions in accurately representing runoff, such as catchments with low runoff mean and variance. The model's performance was evaluated in catchments with minimal human impact, such as dam operations and surface irrigation, to reduce

anthropogenic influences on the results. However, the model did not incorporate detailed land use information beyond basic surface classifications, thereby not accounting for direct human alterations in the hydrological system.

A limitation in our approach was the reliance solely on air temperature and precipitation data for long-term reconstruction, excluding other meteorological factors like sunshine hours, which can only be implicitly approximated by the model via the available input variables. The assumption of static variables, such as land use and glacier coverage, being constant over time

is a necessary simplification but introduces potential inaccuracies. This is particularly critical as land use can vary, and glacier areas are known to decrease over time, potentially leading to biases, especially in the early stages of the reconstruction where observational data are sparse.

Moreover, the dataset used for training the model and the dataset for reconstruction are not entirely independent, though they are not identical. The temporal overlap of the training set within the reconstruction period was unavoidable due to data

limitations. Efforts were made to mitigate the risk of overfitting by employing a distinct validation set, both spatially and temporally separated from the training data.

In runoff modeling, the quality of meteorological drivers has a large impact on model performance, and both meteorological products used here have known limitations. The TabsD product of air temperature shows a clear relationship between the error and the number of stations used for the interpolation, which results in larger errors in the 1960s and 70s most pronounced in

winter months and particularly in the Alps and in Ticino. The linear trend (1961-2010) of interpolated air temperature shows relatively low agreement with the observed trends (Frei, 2014). The RhiresD precipitation product is affected by two primary sources of uncertainty: The rain-gauge measurements are prone to undercatch, leading to underestimation of precipitation particularly with heavy winds and snow in general (Neff, 1977). This leads, in Switzerland, to an underestimation of about 4 % at low elevations and up to 40 % in high altitudes in winter (Sevruk, 1985). From the interpolation, there is a tendency

to overestimate light and underestimate heavy precipitation (MeteoSwiss, 2021b), although these inaccuracies are reduced for areal aggregates such as the catchment averages deployed in the present study. Although no information on the accuracy over time was found, it is expected that the sparser measurement network in the 1960s and 1970s leads to larger errors during this period, similar to the TabsD product. These uncertainties are anticipated to affect the results substantially. We acknowledge that in the early reconstruction period (1960s and 1970s), where fewer measurement stations were available, the reconstruction may

be less trustworthy. The low agreement of interpolated air temperature trends with observations could explain why both the PREVAH and CH-RUN struggle to represent extreme runoff trends. While we did not specifically investigate the representation of extreme runoff events in this study, we expect that the overestimation of low and underestimation of strong precipitation events results in a bias in runoff simulations.

Finally, our deep learning model heavily depends on the availability and diversity of data. Representing infrequent occur-

495 rences or events, which are less common in the data distribution, poses a significant challenge. Consequently, the model's ability to accurately depict rare and extreme hydrological events, such as sudden heavy rains leading to flash floods, is likely limited. This aspect is underscored by the inherent difficulties in modeling the "long tail" of event distribution (Zhang et al., 2023).

# 6   Conclusions

In this study, we developed a data-driven daily runoff reconstruction product for Switzerland, spanning from 1962 to 2023. Our model not only matched but also surpassed the performance of an operational hydrological model at the catchment level. This achievement is particularly noteworthy considering the reduced data requirements, a limitation necessary to achieve such an extensive reconstruction period. Our model effectively captured daily runoff patterns and interannual variability, and represents long-term trends decently, providing a comprehensive and satisfying depiction of runoff dynamics.

The reconstruction product revealed interesting patterns in long-term runoff trends that align with prior knowledge. The additional reconstruction of the 1960s and 1970s suggests that the negative decadal runoff trend is driven by an increase in the frequency, rather than the amplitude, of dry years, along with a decrease in the frequency of wet years. We diagnosed a trend towards lower runoff at the national scale, which was mainly linked to the summer months, where the spatial patterns of runoff indicated increasingly dry conditions particularly in mid-to-high altitudes. We encourage the in-depth investigation of the identified patterns in subsequent studies.

One of the major strengths of our approach lies in its computational efficiency, which opens up possibilities for contiguous near real-time monitoring and potentially forecasting of runoff. The reduced data demands of our model make it an invaluable tool for scenario simulation and attributing trends to anthropogenic climate change, allowing for the rapid evaluation of thousands of scenarios that were not feasible with traditional physically-based models.

Looking ahead, we believe that the current approach could be further enhanced by integrating additional data constraints or incorporating physical knowledge. Specifically, for a more accurate representation of large catchments, we see the inclusion of routing processes as a vital next step.

*Code and data availability.* The code is shared on GitHub and available at https://github.com/bask0/mach-flow/. The upscaling product ("CH-RUN"), including the input data (precipitation, temperature, and the static variables), the PREVAH simulations, and the catchment polygons, is available at https://doi.org/20.500.11850/714281.

## Appendix A: Model training

### A1 Hyperparameter tuning

The hyperparameter space searched is shown in Tab. A1. The `model_dim` denotes the model dimensionality, i.e., the size of the internal representations. The `enc_dropout` refers to dropout (random deactivation of nodes with probability $p$ during training) applied in the encoding layers, and `fusion_method` to the method used for fusion of the temporal and static variables. For the AdamW optimizer, `learning_rate` denotes the step size, and `weight_decay` the L2 regularization. The `temp_layers` refers to the number of stacked temporal layers for both the LSTM and the TCN, and `kernel_size` is the dimensionality of the 1D kernel used for convolution in the time dimension for the latter. An optional `temporal_dropout` is used for the TCN. The performance of the models and the corresponding hyperparameters are provided in Tab. A2 for the LSTMs, and Tab. A3 for the TCNs.

**Table A1.** The search space for hyperparameter tuning. The common hyperparameters were used for both architectures, the other ones are model-specific.

| Name | Search space |
|---|---|
| **Common parameters** | |
| `model_dim` | {64, 128, 256 } |
| `enc_dropout` | {0.0, 0.2} |
| `fusion_method` | {'pre_encoded', 'pre_repeated', 'post_repeated'} |
| `learning_rate` | {1e-4, 1e-3, 1e-2} |
| `weight_decay` | {1e-1, 1e-2, 1e-3} |
| **LSTM parameters** | |
| `temp_layers` | {1, 2} |
| **TCN parameters** | |
| `temp_layers` | {2, 3, 4} |
| `kernel_size` | {8, 16} |
| `temporal_dropout` | {0.0, 0.2} |

The `model_dim` corresponds to the model dimensionality, which is shared among all feed-forward and temporal neural networks. `enc_dropout` was used in the encoder layers prior to the temporal layer, and `fusion_method` corresponds to the approaches described in Sect. 3.1. `learning_rate` and `weight_decay` are parameters of the optimizer and control the weight update step size and regularization, respectively. For the long short-term memory (LSTM) and the temporal convolutional network (TCN), `temp_layers` defines the number of stacked temporal layers. For the latter, `kernel_size` is the width of the 1D convolution kernel applied along the time dimension, and `temporal_dropout` deactivates entire channels of input encoding, instead of randomly dropping activations.

| Rank | NSE | allbasins | sqrttrans | static | model_dim | enc_dropout | fusion_method | temp_layers | learning_rate | weight_decay | num_params |
|---|---|---|---|---|---|---|---|---|---|---|---|
| 1 | 0.76 | True | True | all | 128 | 0.2 | pre_encoded | 1 | 0.001 | 0.1 | 169 K |
| 2 | 0.74 | False | True | all | 256 | 0.2 | pre_encoded | 2 | 0.001 | 0.01 | 1200 K |
| 3 | 0.72 | False | False | dred | 128 | 0.2 | pre_encoded | 2 | 0.001 | 0.1 | 298 K |
| 4 | 0.72 | True | False | all | 256 | 0.2 | pre_encoded | 1 | 0.001 | 0.01 | 666 K |
| 7 | 0.71 | False | False | all | 128 | 0.2 | pre_encoded | 2 | 0.01 | 0.1 | 301 K |
| 8 | 0.71 | True | True | dred | 128 | 0.2 | pre_encoded | 2 | 0.001 | 0.1 | 298 K |
| 10 | 0.70 | False | True | area | 128 | 0.0 | post_repeated | 2 | 0.001 | 0.1 | 330 K |
| 11 | 0.70 | True | True | area | 256 | 0.0 | post_repeated | 1 | 0.001 | 0.01 | 790 K |
| 13 | 0.69 | True | False | area | 256 | 0.2 | pre_repeated | 1 | 0.0001 | 0.01 | 659 K |
| 14 | 0.69 | False | True | dred | 256 | 0.2 | pre_encoded | 2 | 0.001 | 0.001 | 1200 K |
| 15 | 0.68 | False | False | area | 64 | 0.0 | post_repeated | 1 | 0.001 | 0.1 | 50 K |
| 18 | 0.66 | True | False | dred | 128 | 0.2 | pre_encoded | 2 | 0.01 | 0.1 | 298 K |

**Table A2.** Hyperparameters found by tuning for the LSTM-based architectures. The rows are sorted by the catchment-level Nash-Sutcliffe modeling efficiency (NSE) in descending order, and the rank column represents the overall rank among all models. White columns denote the model setup. Yellow columns represent architecture, the magenta ones optimizer parameters, and cyan is the number of tunable parameters. The column `allbasins` (using additional catchments for training or not), `sqrttrans` (transform runoff with square root or not), and `static` (use all static variable, a dimensionality-reduced version, or none) refer to the factorial experiments described in Sect. 3.5.

| Rank | NSE | allbasins | sqrttrans | static | model_dim | enc_dropout | fusion_method | temp_layers | kernel_size | learning_rate | weight_decay | num_params |
|---|---|---|---|---|---|---|---|---|---|---|---|---|
| 5 | 0.72 | True | False | dred | 128 | 0.0 | post_repeated | 3 | 16 | 0.0001 | 0.001 | 1600 K |
| 6 | 0.72 | False | True | dred | 128 | 0.0 | post_repeated | 4 | 16 | 0.0001 | 0.001 | 2200 K |
| 9 | 0.71 | True | True | all | 256 | 0.0 | post_repeated | 4 | 8 | 0.0001 | 0.1 | 4500 K |
| 12 | 0.70 | True | True | dred | 128 | 0.0 | post_repeated | 4 | 16 | 0.0001 | 0.001 | 2200 K |
| 16 | 0.68 | False | True | area | 64 | 0.0 | post_repeated | 4 | 16 | 0.001 | 0.1 | 542 K |
| 17 | 0.68 | True | False | area | 128 | 0.2 | pre_repeated | 3 | 16 | 0.0001 | 0.1 | 1600 K |
| 19 | 0.66 | False | False | area | 128 | 0.2 | post_repeated | 4 | 8 | 0.001 | 0.1 | 1100 K |
| 20 | 0.65 | True | True | area | 128 | 0.0 | post_repeated | 4 | 16 | 0.0001 | 0.001 | 2200 K |
| 21 | 0.65 | False | True | all | 256 | 0.2 | post_repeated | 4 | 16 | 0.0001 | 0.1 | 8700 K |
| 22 | 0.64 | False | False | all | 256 | 0.2 | post_repeated | 4 | 8 | 0.001 | 0.01 | 4500 K |
| 23 | 0.63 | False | False | dred | 64 | 0.0 | pre_repeated | 4 | 8 | 0.001 | 0.01 | 272 K |
| 24 | 0.62 | True | False | all | 256 | 0.2 | pre_encoded | 4 | 8 | 0.0001 | 0.1 | 4300 K |

**Table A3.** Hyperparameters found by tuning for the TCN-based architectures. The rows are sorted by the catchment-level Nash-Sutcliffe modeling efficiency (NSE) in descending order, and the rank column represents the overall rank among all models. White columns denote the model setup. Yellow columns represent architecture, the magenta ones optimizer parameters, and cyan is the number of tunable parameters. The column `allbasins` (using additional catchments for training or not), `sqrttrans` (transform runoff with square root or not), and `static` (use all static variable, a dimensionality-reduced version, or none) refer to the factorial experiments described in Sect. 3.5.

From the fusion methods introduced in Sect. 3.1, prefusion with encoding was selected most often for the LSTM architecture, while postfusion was more commonly selected for the TCN (A, Tab. A2 & A3). Note that the fusion was part of the hyperparameter tuning, and only the best approach is used to make the final predictions. For both architectures, prefusion with encoding was commonly selected if all static variables were used as input. Interestingly, as seen in the Tab. A2 & A3, the number of tunable parameters, an outcome of the hyperparameter tuning process, was larger by a factor of five for the TCN architectures (2.8 million in average) compared to the LSTMs (0.52 million in average).

## A2 Comparison of model setups

Here, we evaluate the factorial experiment outlined in Sect. 3.5. In Fig. A1, the diagonal shows how the model setups impact the median NSE across catchments, and the offset triangular shows the interactions of the factors.

For the LSTM architecture, using all training catchments and all static variables had a larger impact on the outcome than the square root transform of the target variable, which was negligible. The factors "training catchments" and "static variables" interacted strongly, indicating that having both more training data and more information on catchment properties contributes more to the model performance than using the factors independently. The interaction with "target transform", in contrast, was

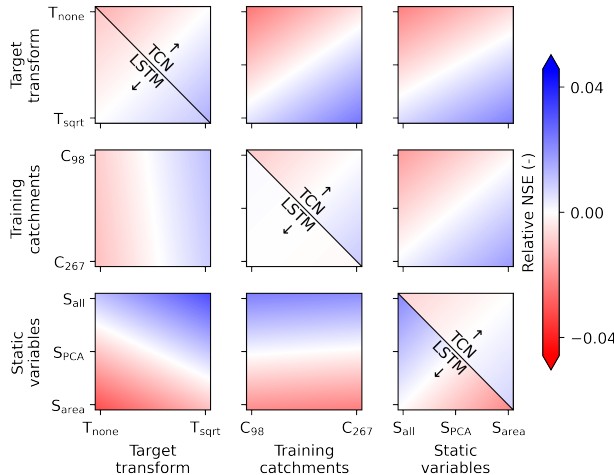

**Figure A1.** Model setup impact on performance and their interactions based on the median catchment Nash-Sutcliffe modeling efficiency (NSE), evaluated on the spatially and temporally independent test set. The colors represent NSE relative to the respective model mean across setups; red represents worse, and blue represents better performance. The background gradients have been calculated using linear least square regression with the model performance as dependent variable. The offset lower triangular panels show results for the LSTM, the offset upper triangular for the TCN model, i.e., variable interactions, while the three panels on the diagonal show variable main effects, split to distinguish between the two models. The x and y axes represent the setups tested in the factorial experiment. *Training catchments* refers to the catchments used for training: a subset of 98 catchments less impacted by anthropogenic factors ($C_{98}$), or all catchments ($C_{267}$). *Target transform* is either $T_{none}$ if the target variable was not transformed, or $T_{sqrt}$ for square root transform. *Static variables* is $S_{all}$ if all static variables are used, $S_{PCA}$ if they are first transformed using PCA, or $S_{area}$ if only catchment area is used beyond the meteorological variables.

minimal for the LSTM architectures. For the TCN models, the results look different. Using more static variables as input seems to have improved model performance, while using additional training catchments did so only marginally, and the interactions of the "training catchments" and "static variables" was less clear. Overall, the TCN models show lower range of performance across setups than the LSTM models.

*Author contributions.* The study was planned and designed by BK, LG, MS, and WA. Data was processed by MS and WA. MZ provided the PREVAH benchmark. The data pipeline and model implementation, and visualization of results were conducted by BK. BK took the lead in the preparation of the manuscript, but all authors contributed.

*Competing interests.* The contact author has declared that none of the authors has any competing interests.

*Acknowledgements.* We acknowledge the funding from the Swiss Data Science Center project C20-02 *Machine learning for Swiss (CH) river flow estimation (MACH-Flow)*. We would like to express our gratitude to the anonymous reviewer #1 and Ross Woods, and Editor Nunzio Romano for their thoughtful comments, constructive feedback, and guidance throughout the peer-review process, which significantly improved the quality of this manuscript.

555

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
