# Peer review of "CH-RUN: A deep-learning-based spatially contiguous runoff reconstruction for Switzerland"

_EGUsphere, 2024_

## Referee Comment (RC2)

CH-RUN: A data-driven spatially contiguous runoff monitoring product for Switzerland

**Contribution**

The manuscript presents a method for creating spatially and temporally complete reconstructions of historical runoff for Switzerland, using machine learning techniques. The model can be run at low computational cost. The results compare favourably with reconstructions created using a complex distributed hydrological model over the same domain. Simulated catchment runoff data demonstrate temporal trends at decadal scales which are consistent with those previously reported.

**Assessment**

I do not have any major concerns about the paper. In general it is clearly written, and logically consistent, with appropriate caveats. I do not have expertise in current machine learning methods, and can't comment usefully on the methods employed. I do have are a number of concerns listed below, but none of them are likely to undermine the general conclusion of the paper (though see my major comment about robust quantification of trends).

**Major Comments**

1. L287 "Figure 5c illustrates how the models capture spatial patterns of annual trends" I'm not confident that the results in Fig 5c are robust. First, since there is no evidence shown that temporal trends for individual catchments are linear, it would be prudent to use a non-parametric trend slope (e.g., Sen slope), rather than a linear fit for each catchment. Second, the PREVAH points in Fig 5c look to be mostly a cloud, and it seems possible that the two points in the upper right have a lot of influence in determining the regression line. I suggest the authors either demonstrate that those two points are not influential by recomputing the regression without them and showing that it does not change much, or else use a robust regression method. Similar comments apply to the influence of the single LSTM point in the upper right. The use of a Pearson correlation coefficient to describe the association between variables in Fig 5c seems similarly unjustified, and Spearman rank correlations would be more appropriate in the presence of potential outliers.
I list this as a major comment because the paper includes in its conclusion (L464) the claim that "Our model effectively captured ... long-term trends"

**Minor Points**

2. L2. I would like to read slightly more detail about the method in the abstract. A few additional key words to let the reader know which specific machine learning technique(s) was preferred.

3. L10. "capturing annual variability" it would be clearer to say inter-annual, rather than annual.

4. L12. "These are characterized by an increased occurrence of dry years, contributing to a negative decadal trend, particularly during the summer months." A negative trend in what?

5. L16. "the reduced data dependency … of our model" This statement makes sense in the context of comparison against PREVAH, but there are other traditional hydrological models which only require temperature and precipitation (e.g., HBV) which also have reduced data dependency; this is not a feature that is unique to these machine learning models.

6. L54 "2.2 Meteorological drivers" What is the spatial resolution of the gridded products? What is known about their accuracy, in particular, at high altitudes? How does the time-varying availability of the underlying climate observations affect the reliability of the product, especially in the early part of the period when presumably fewer stations are available?

7. Section 3: I do not have expertise in current machine learning methods, and can't comment usefully on most of this section. The model evaluation appears to be well designed.

8. L258 "Understanding the capabilities of our model necessitates a thorough evaluation of daily runoff simulations" This sentence indicates that a thorough evaluation is about to be presented. However, I think that overstates the analysis which follows. Analysing model outputs in terms of squared differences between the measured and modelled time series, and then annual means, is a useful, but limited evaluation. There are many other ways to assess the performance of a model (e.g., its ability to reproduce multiple hydrologic signatures of interest). Does the model reproduce flood peaks well? Low flows? Seasonal variation? Recession characteristics? I think that the analysis provided is appropriate for this paper, but it's a stretch to call it a "thorough evaluation", so the phrase should be modified slightly.

9. Sections 4.2 and 4.3: I found the material here very helpful and well presented.

10. L352 "under these data-limited conditions" The point about being data limited is made several times. Can you explain why you say that having 98 catchments is data-limited? The spatial coverage of Switzerland is clearly patchy and partial (Figure 1), but that wouldn't matter if spatial correlation lengths were large. Is there an objective method for assessing the extent to which any streamflow dataset contains a large or small amount of information, relative to a space (and time) domain of interest?

11. L418 "We hypothesize that the negative trend in summer is less related to snowmelt but rather connected to an increase in evapotranspiration via warmer air temperatures," It

still seems possible to me that changes in snowmelt might affect summer streamflow. For what reason do you prefer your alternative explanation? How might such a hypothesis be tested to discriminate between these potential causes? I think that proposing hypotheses in a discussion is a great idea, but it would be good to know that they were testable, at least in principle.

12. L468 "… is contributing towards the negative decadal trend." Negative decadal trend in what?

13. L469 "… and linked to the summer months" This phrase is vague, and could be made much more specific. What happened in summer? Is it a cause or effect of the drier conditions?

14. Table A1 caption "onse" should be "ones"

15. L486 deontes should be denotes

16. L488 temporal_dropoput should be temporal_dropout

17. Table A2 caption. The meanings of the columns "allbasins sqrttrans static" are not defined; please refer the reader to the relevant material in section 3

---

## Author Comment (AC1)

**CH-RUN: A data-driven spatially contiguous runoff monitoring product for Switzerland; https://doi.org/10.5194/egusphere-2024-993**

**Response to anonymous reviewer #1**; https://doi.org/10.5194/egusphere-2024-993-RC1

*Comment 1*

*My first comment is about the paper title. I expected a somewhat different study based on the title. The mentioning of a runoff monitoring product in the title suggests some type of derived data product, though the focus of the manuscript is the extensive development of neural network approaches to perform the reconstruction. I assume that this is when the authors say data-driven, though I do think a title that better represents the actual study content would be preferable.*

We agree that the title could be misleading and decided to change it to: "CH-RUN: A deep-learning-based spatially contiguous runoff reconstruction for Switzerland"

*Comment 2 and 3*

*[2] The authors' view of "traditional hydrological models" is overly narrow (lines 24ff.). While physically-based (pb) models, like the one previously developed for Switzerland (PREVAH), have a high computational demand and are rather data hungry, this is not the case for all hydrological models. In fact, much of hydrology uses rather parsimonious models (GR4J, HyMod, PDM…) which do not put a high demand on computational resources. It would be good if the authors either refine their statement to pb models or widen it to include a wider range of model complexities. Given that the simulation of daily runoff is done with such simpler models in many countries, I would suggest the latter.*

*[3] A similar point can be made about the data need of hydrological models which is discussed in lines 36ff. Several widely used hydrological models can be driven by precipitation and temperature only – if they are of the parsimonious type*

Overall, we agree with these comments. There are indeed less computationally demanding and data-hungry hydrological models. Such fast models are usually calibrated per catchment, and then regionalized (and so is PREVAH). While the specific data requirements and processing speed differs vastly among models, the end-to-end deep learning-based approach used here encompasses all of the mentioned advantages. Compared to PREVAH, which is used widely in scientific context to run scenarios and projections, our approach is fast and data-efficient.

We added a short note on different types of hydrological models to the introduction:

Traditional hydrological models offer pivotal insights into land-surface processes. For Switzerland, a diverse array of hydrological models has been employed (Horton et al., 2022), ranging from complex ones, which are heavily founded on physical principles, to lightweight ones using conceptual process representations with calibrated parameters. While the former offer detailed insights and control, they rely on a large number of inputs and are computationally expensive. The latter, in contrast, can be parsimonious in terms of data and computational resources, yet they need to be calibrated per catchment, which limits their applicability to prediction in ungauged catchments. The generalization to ungauged catchments via regionalization is possible, but introduces another layer of complexity (Beck et al., 2016). As a complementary approach, deep learning holds potential as a tool for hydrological modeling, both in terms of performance and efficiency (Nearing et al., 2021), and it comes with built-in regionalization when trained on multiple catchments jointly (Kratzert et al., 2024).

With this paragraph, we wanted to calrify that process-based models can be fast and data-efficient as well. We also made the following changes:

Instead of "reduced data requirements", we now write "low data requirements".

Furthermore, the low data requirements and computational efficiency of our model pave the way for simulating diverse scenarios and conducting comprehensive climate attribution studies. This represents a substantial progression in the field, allowing for the analysis of thousands of scenarios in a time frame significantly shorter than traditional methods.

We added "reduced data needs [...] *compared to the PREVAH model*."

The reconstruction of runoff back to the early 1960s for Switzerland is a novelty enabled by the reduced data needs of our deep learning-based approach compared to the PREVAH model. Here, we evaluate the plausibility of the simulated patterns based on Fig. 7-9 by contrasting them to prior knowledge.

The overall trend towards drier conditions simulated by our data-driven model aligns with independent studies. This has

**Comment 4**

*The authors use squared error metrics for model calibration. They then use disaggregated components of such metrics for further analysis- which I like. What I missed in the analysis is any assessment of whether these components show any structure across Switzerland. For example, Gudmundsson et al. (2012, WRR) showed for example a strong correlation between bias errors and elevation differences for some comparable catchments to those used here. Did you look for any systematic biases (in the context of Fig. 4)?*

We agree that such an analysis would be interesting and decided to add a figure on spatial errors (new Figure 4, see below). With this figure, we can discuss spatial patterns of error components and link them to catchment properties. We also added the difference in the error components (LSTM minus PREVAH), to better understand how the models compare on a spatial basis. We also performed an explorative analysis to better understand the relationship between model performance (difference) and catchment properties.

The new figure:

[revised manuscript text omitted]

290    catchments that also had low runoff variance.

**Comment 5**

*The relative NSE range shown in the legend of Figure 3 seems very small. Is the variability shown in the various small plots actually relevant?*

We will move this figure and its discussion to the appendix to make space for the more relevant figure from comment [4].

**Comment 6**

*The other reviewers already made some comments and suggestions regarding the trend analysis performed, and the need to test a non-parametric strategy. I will not repeat his points in this review, but I believe that they are justified.*

We agree with this criticism of the trend analysis. We followed the suggestion of the other reviewer and used Sen's slope to compute the robust trend per catchment. To reduce the impact of outliers for the comparison of observed and simulated trends, we use robust estimates for regression and Spearman's rank correlation.

With this robust analysis, the difference between CH-RUN and PREVAH are slightly reduced, but still, CH-RUN reproduces linear trends better with rank correlation $\rho=0.6$, compared to PREVAH with $\rho=0.42$. Both models appear to reproduce the trends less accurately compared to the previous analysis.

The updated results:

[Figure]

**Figure 5.** Catchment-level evaluation at the annual scale. a) The Pearson correlation (r) and b) bias in $\mathrm{mm\,y^{-1}}$ distribution across 98 training catchments evaluated on the test set. c) The simulated annual runoff trends compared to observations. The points represent the linear trend (found by robustified least squares fit) of single catchments. Note that for the trend calculation, the time range from 1995 (start of first test period) to 2020 (end of second test period) was used. The inset equation shows the linear least square fit and the corresponding rank correlations.

For a reconstruction product, it is crucial to adequately represent yearly variability and long-term trends. We, therefore, evaluate this aspect on annual runoff aggregates (Fig. 5). The best-performing model, $\mathrm{LSTM_{best}}$, represented the interannual variability (Fig. 5a), quantified as the Pearson correlation coefficient between the annual values for each catchment, well with a median of $r = 0.93$, and with 75 % of catchments above $r = 0.85$. The bias averages close to zero and for 50 % of the catchments, it was in the range of -250 to 250 $\mathrm{mm\,y^{-1}}$ (Fig. 5b). On the interannual variability, PREVAH showed a slightly lower correlation (Fig. 5a) across catchments with a median of $r = 0.91$. In terms of bias, PREVAH performed marginally better with a median closer to zero and a lower spread (Fig. 5b).

Figure 5c illustrates how the models captured spatial patterns of annual trends between January 1995 and December 2020 (Fig. 5c). The agreement was calculated by first computing the catchment-level linear trends for the observations and the simulations by PREVAH and $\mathrm{LSTM_{best}}$ independently using the robust Theil-Sen estimator (Sen, 1968). Then, we fit a regression between the observed and estimated trend slopes by the two models using robust regression with Huber weighting and the default tuning constant of $c = 1.345$ (Huber and Ronchetti, 2009). This approach reduces the impact of outliers by giving lower weight to large residuals. For quantifying the alignment of the simulated trends, we us Spearman correlation ($\rho$), which is relatively robust against outliers. While the $\mathrm{LSTM_{best}}$ represented the spatial patterns of the linear trend relatively well with a correlation of $\rho = 0.60$, PREVAH achieveed a correlation of $\rho = 0.42$. Both models underestimated the strength of negative and positive trends with slopes of 0.52 ($\mathrm{LSTM_{best}}$) and 0.64 (PREVAH), and they exhibited small negative biases of -3.73 ($\mathrm{LSTM_{best}}$) and -6.44 (PREVAH) $\mathrm{mm\,y^{-1}}$.

**Comment 7**

*Rather than the qualitative evaluation in section 4.2, is there not enough information in the 98 catchment differences to show where and when PREVAH is better/worse?*

See answer to comment [4].

**Comment 8**

*Section 5.3 "The reconstruction of runoff back to the early 1960s for Switzerland is a novelty enabled by the reduced data needs of our deep learning-based approach." But would the NN benefit from additional data?*

Yes, the neural networks would benefit from additional covariates. We did not systematically test this, but in preliminary model runs we found that adding more covariates (radiation, vp, tmin, tmax) helped. The differences were, however, not substantial. The dependency on air temperature and precipitation alone allowed us to extend the reconstruction back to the 1960s, which enabled the monitoring of long-term trends. Testing the capabilities of deep learning approaches in the time domain in more data-abundant periods has been done before and is out of the scope of this study.

**Comment 9**

*(lines 446ff.) The authors state that "A limitation in our approach was the reliance solely on air temperature and precipitation data for long-term reconstruction, excluding other meteorological factors like cloud-related effects, which could only be indirectly approximated by the model." Can you name examples of hydrological models that consider cloud-related effects? Do you mean the consideration of sunshine hours? You could have used such information, couldn't you?*

Yes, we meant cloud effects on radiation. The latter is used in many hydrological models (e.g., SWBM, PREVAH). The deep learning models can, in principle, learn such effects implicitly (precipitation means clouds means less radiation). We could use such data, yes, but sunshine hours are only available from the 1970s from MeteoSwiss. We changed the wording and hope that it is clearer now:

> A limitation in our approach was the reliance solely on air temperature and precipitation data for long-term reconstruction, excluding other meteorological factors like sunshine hours, which can only be implicitly approximated by the model via the available input variables. The assumption of static variables, such as land use and glacier coverage, being constant over time
> 470 is a necessary simplification but introduces potential inaccuracies. This is particularly critical as land use can vary and glacier areas are known to decrease over time, potentially leading to biases, especially in the early stages of the reconstruction where observational data are sparse.

**Comment 10**

*Also, the authors state that the "The assumption of static variables, such as land use and glacier coverage, being constant over time is a necessary simplification but introduces potential inaccuracies." I am not completely clear why this is a necessary simplification. Why can changing forest cover and a limited contribution of melting glaciers not be included?*

In principle, we could use such data, but it is difficult to find high-quality and harmonized historical data on land use or glaciers covering the entire period back to the 1960s. Even if such data is available in some form, the model architecture would need to be adapted to deal with inconsistent resolution etc. Thus, we consider this an interesting suggestion, but unfortunately out of the scope of this study.

**Comment 11**

*Conclusions: "One of the major strengths of our approach lies in its computational efficiency, which opens up possibilities for contiguous near real-time monitoring and potentially forecasting of runoff." And "…allowing for the rapid evaluation of thousands of scenarios that were not feasible with traditional physically-based models." Here the authors state their assumption of "traditional physically-based models" which is not the same as traditional hydrological models. It would be good to clarify this difference in the Introduction section.*

This should read "traditional hydrological models". We will clarify this in the revised manuscript. The broader discussion about model types and their strengths and weaknesses is covered in the answers to comment 3 and 4.

---

## Author Comment (AC2)

**CH-RUN: A data-driven spatially contiguous runoff monitoring product for Switzerland; https://doi.org/10.5194/egusphere-2024-993**

**Response to reviewer #2 (Ross Woods);** https://doi.org/10.5194/egusphere-2024-993-RC2

*Comment 1*

*L287 "Figure 5c illustrates how the models capture spatial patterns of annual trends". I'm not confident that the results in Fig 5c are robust. First, since there is no evidence shown that temporal trends for individual catchments are linear, it would be prudent to use a non-parametric trend slope (e.g., Sen slope), rather than a linear fit for each catchment. Second, the PREVAH points in Fig 5c look to be mostly a cloud, and it seems possible that the two points in the upper right have a lot of influence in determining the regression line. I suggest the authors either demonstrate that those two points are not influential by recomputing the regression without them and showing that it does not change much, or else use a robust regression method. Similar comments apply to the influence of the single LSTM point in the upper right. The use of a Pearson correlation coefficient to describe the association between variables in Fig 5c seems similarly unjustified, and Spearman rank correlations would be more appropriate in the presence of potential outliers. I list this as a major comment because the paper includes in its conclusion (L464) the claim that "Our model effectively captured ... long-term trends"*

We agree with this criticism of our trend analysis. We followed your suggestion and used Sen's slope to compute the robust trend per catchment. To reduce the impact of outliers for the comparison of observed and simulated trends, we use robust estimates for regression and Spearman's rank correlation.

With this robust analysis, the difference between CH-RUN and PREVAH are slightly reduced, but still, CH-RUN reproduces linear trends better with rank correlation $\rho=0.6$, compared to PREVAH with $\rho=0.42$. Both models appear to reproduce the trends less accurately compared to the previous analysis.

The updated results:

[Figure]

**Figure 5.** Catchment-level evaluation at the annual scale. a) The Pearson correlation (r) and b) bias in $\mathrm{mm\,y^{-1}}$ distribution across 98 training catchments evaluated on the test set. c) The simulated annual runoff trends compared to observations. The points represent the linear trend (found by robustified least squares fit) of single catchments. Note that for the trend calculation, the time range from 1995 (start of first test period) to 2020 (end of second test period) was used. The inset equation shows the linear least square fit and the corresponding rank correlations.

For a reconstruction product, it is crucial to adequately represent yearly variability and long-term trends. We, therefore, evaluate this aspect on annual runoff aggregates (Fig. 5). The best-performing model, $\mathrm{LSTM_{best}}$, represented the interannual variability (Fig. 5a), quantified as the Pearson correlation coefficient between the annual values for each catchment, well with a median of $r = 0.93$, and with 75 % of catchments above $r = 0.85$. The bias averages close to zero and for 50 % of the catchments, it was in the range of -250 to 250 $\mathrm{mm\,y^{-1}}$ (Fig. 5b). On the interannual variability, PREVAH showed a slightly lower correlation (Fig. 5a) across catchments with a median of $r = 0.91$. In terms of bias, PREVAH performed marginally better with a median closer to zero and a lower spread (Fig. 5b).

Figure 5c illustrates how the models captured spatial patterns of annual trends between January 1995 and December 2020 (Fig. 5c). The agreement was calculated by first computing the catchment-level linear trends for the observations and the simulations by PREVAH and $\mathrm{LSTM_{best}}$ independently using the robust Theil-Sen estimator (Sen, 1968). Then, we fit a regression between the observed and estimated trend slopes by the two models using robust regression with Huber weighting and the default tuning constant of $c = 1.345$ (Huber and Ronchetti, 2009). This approach reduces the impact of outliers by giving lower weight to large residuals. For quantifying the alignment of the simulated trends, we us Spearman correlation ($\rho$), which is relatively robust against outliers. While the $\mathrm{LSTM_{best}}$ represented the spatial patterns of the linear trend relatively well with a correlation of $\rho = 0.60$, PREVAH achieveed a correlation of $\rho = 0.42$. Both models underestimated the strength of negative and positive trends with slopes of 0.52 ($\mathrm{LSTM_{best}}$) and 0.64 (PREVAH), and they exhibited small negative biases of -3.73 ($\mathrm{LSTM_{best}}$) and -6.44 (PREVAH) $\mathrm{mm\,y^{-1}}$.

Also, we changed the statement "Our model effectively captured ... long-term trends":

**6 Conclusions**

In this study, we developed a data-driven daily runoff reconstruction product for Switzerland, spanning from 1962 to 2023. Our model not only matched but also surpassed the performance of an operational hydrological model at the catchment level. This achievement is particularly noteworthy considering the reduced data requirements, a limitation necessary to achieve such an extensive reconstruction period. Our model effectively captured daily runoff patterns and interannual variability, and represents long-term trends decently, providing a comprehensive and satisfying depiction of runoff dynamics.

**Comment 2**
*[2] L2. I would like to read slightly more detail about the method in the abstract. A few additional key words to let the reader know which specific machine learning technique(s) was preferred.*

Indeed, the abstract did not provide much information about the methods used. We will add more details on the machine learning techniques used in the revised version.

The updated abstract:

**Abstract.**

   This study presents a data-driven reconstruction of daily runoff that covers the entirety of Switzerland over an extensive period from 1962 to 2023. To this end, we harness the capabilities of deep learning-based models to learn complex runoff-generating processes directly from observations, thereby facilitating efficient large-scale simulation of runoff rates at ungauged

5    locations. We test two sequential deep learning architectures, a long short-term memory (LSTM) model, a recurrent neural network able to learn complex temporal features from sequences, and a convolution-based model, which learns temporal dependencies via 1D convolutions in the time domain. The models receive temperature, precipitation, and static catchment properties as input. By driving the resulting model with gridded temperature and precipitation data available since the 1960s, we provide a spatiotemporally continuous reconstruction of runoff. The efficacy of the developed model is thoroughly as-

10   sessed through spatiotemporal cross-validation and compared against a distributed hydrological model used operationally in Switzerland.

**Comment 3**

*[3] L10. "capturing annual variability" it would be clearer to say inter-annual, rather than annual.*

This suggestion will be adopted in the revised manuscript

**Comment 4**

*L12. "These are characterized by an increased occurrence of dry years, contributing to a negative decadal trend, particularly during the summer months." A negative trend in what?*

A negative decadal trend in runoff. This will be clarified in the revised version.

**Comment 5**

*L16. "the reduced data dependency ... of our model" This statement makes sense in the context of comparison against PREVAH, but there are other traditional hydrological models which only require temperature and precipitation (e.g., HBV) which also have reduced data dependency; this is not a feature that is unique to these machine learning models.*

We agree with this comment. There are hydrological models with reduced data dependency and fast computation. We will avoid such a general statement and instead focus on the comparison to PREVAH, which is a state-of-the-art model used operationally in Switzerland. Our approach has a built-in regionalization and performs reasonably well and fast, and has reduced data needs compared to PREVAH.

We changed "reduced data dependency" to "low data requirements":

   Furthermore, the low data requirements and computational efficiency of our model pave the way for simulating diverse

20   scenarios and conducting comprehensive climate attribution studies. This represents a substantial progression in the field, allowing for the analysis of thousands of scenarios in a time frame significantly shorter than traditional methods.

We also made clear that data requirements are reduced *compared to PREVAH*.

> The reconstruction of runoff back to the early 1960s for Switzerland is a novelty enabled by the reduced data needs of our deep learning-based approach compared to the PREVAH model. Here, we evaluate the plausibility of the simulated patterns based on Fig. 7-9 by contrasting them to prior knowledge.
>
> 425    The overall trend towards drier conditions simulated by our data-driven model aligns with independent studies. This has

**Comment 6**

*L54 "2.2 Meteorological drivers" What is the spatial resolution of the gridded products? What is known about their accuracy, in particular, at high altitudes? How does the time-varying availability of the underlying climate observations affect the reliability of the product, especially in the early part of the period when presumably fewer stations are available?*

The spatial resolution of the meteorological data products (1 km) is now mentioned in the data section.

We added the following paragraph on meteorological data to the limitations section:

> In runoff modeling, the quality of meteorological drivers has a large impact on model performance, and both meteorological products used here have known limitations. The TabsD product of air temperature shows a clear relationship between the error and the number of stations used for the interpolation, which results in larger errors in the 1960s and 70s most expressed in
> 480    winter months and particularly in the Alps and in Ticino. The linear trend (1961-2010) of interpolated air temperature shows relatively low agreement with the observed trends (Frei, 2014). The RhiresD precipitation product is affected by two primary sources of uncertainty: The rain-gauge measurements are prone to undercatch, leading to underestimation of precipitation particularly with heavy winds and snow in general (Neff, 1977). This leads, in Switzerland, to an underestimation of about 4 % at low elevations and up to 40 % in high altitudes in winter (Sevruk, 1985). From the interpolation, there is a tendency
> 485    to overestimate light and underestimate heavy precipitation (MeteoSwiss, 2021b), although these inaccuracies are reduced for areal aggregates such as the catchment averages deployed in the present study. We did not find any information on the accuracy over time, but we expect that the sparser measurement network in the 1960s and 70s leads to larger errors during this period, similar to the TabsD product. We expect that these uncertainties affect our results substantially. We acknowledge that in the early reconstruction period (1960s and 70s), where less measurement stations were available, the reconstruction may
> 490    be less trustworthy. The low agreement of interpolated air temperature trends with observations could explain why both the PREVAH and CH-RUN struggle to represent extreme runoff trends. While we did not specifically investigate the representation of extreme runoff events in this study, we expect that the overestimation of low and underestimation of strong precipitation events results in a bias in runoff simulations.

**Comment 7**

*Section 3: I do not have expertise in current machine learning methods, and can't comment usefully on most of this section. The model evaluation appears to be well designed.*

**Comment 8**

*L258 "Understanding the capabilities of our model necessitates a thorough evaluation of daily runoff simulations" This sentence indicates that a thorough evaluation is about*

*to be presented. However, I think that overstates the analysis which follows. Analyzing model outputs in terms of squared differences between the measured and modeled time series, and then annual means, is a useful, but limited evaluation. There are many other ways to assess the performance of a model (e.g., its ability to reproduce multiple hydrologic signatures of interest). Does the model reproduce flood peaks well? Low flows? Seasonal variation? Recession characteristics? I think that the analysis provided is appropriate for this paper, but it's a stretch to call it a "thorough evaluation", so the phrase should be modified slightly.*

Indeed, the main objective of this paper is the presentation of the methods and the runoff reconstruction product. In this sense, we agree with this comment; our evaluation is a general assessment rather than a thorough one.

The updated part:

**4.1.1 Model performance**

To understand the capabilities of our model to represent daily runoff at catchment level, we evaluate the model performance first. Figure 3 presents the empirical cumulative density functions for different metrics across the 98 catchments. Models based

250  on the TCN architecture are depicted in blue, those using LSTM networks in red, and the PREVAH model is represented in black. The model with the best performance is emphasized using a thicker line. Panel a focuses on the Nash-Sutcliffe Efficiency (NSE), while panels b–d provide a detailed breakdown of the Mean Squared Error (MSE) into its components – squared bias, variance error, and phase error – as previously introduced.

**Comment 9**
*Sections 4.2 and 4.3: I found the material here very helpful and well presented.*

Thank you.

**Comment 10**
*L352 "under these data-limited conditions" The point about being data limited is made several times. Can you explain why you say that having 98 catchments is data-limited? The spatial coverage of Switzerland is clearly patchy and partial (Figure 1), but that wouldn't matter if spatial correlation lengths were large. Is there an objective method for assessing the extent to which any streamflow dataset contains a large or small amount of information, relative to a space (and time) domain of interest?*

Deep neural networks require a large amount of data to be trained, and especially extreme events are scarce by definition. More data would likely increase the performance of both the LSTM and the TCN (this has been shown by others, but is out of scope of this study), and we hypothesize that a model with more parameters (the TCN) could profit more.

The second point about the information content of a streamflow dataset raises a challenging question. A closely related concept is the "area of applicability" (Meyer et al. (2021); https://doi.org/10.1111/2041-210X.13650), which would tell us how well the extrapolation in space is constrained by data. This approach was, however, developed

for static data, and the temporal data we use renders the problem more complex. While we acknowledge the relevance of this question, it falls outside the scope of our study.

**Comment 11**

*L418 "We hypothesize that the negative trend in summer is less related to snowmelt but rather connected to an increase in evapotranspiration via warmer air temperatures," It still seems possible to me that changes in snowmelt might affect summer streamflow. For what reason do you prefer your alternative explanation? How might such a hypothesis be tested to discriminate between these potential causes? I think that proposing hypotheses in a discussion is a great idea, but it would be good to know that they were testable, at least in principle.*

We do agree that this statement was not supported by our analysis and we will change it in the revised version of the manuscript. The earlier snowmelt could indeed cause lower runoff in the summer months. The main objective of this study is the presentation of a new data product, accompanied by some basic sanity checks. We encourage the testing of such hypotheses by follow-up studies.

The updated part:

> contribution to negative trends in later summer. In Ticino, a strong trend towards warmer temperatures has been reported, although precipitation seems to not show significant trends (Reinhard et al., 2005). The negative trend in summer is likely
>
> 440  caused by both a lack of snowmelt and an increase in evapotranspiration via warmer air temperatures, which can have a significant impact on runoff (Teuling et al., 2013; Goulden and Bales, 2014).

**Comment 12**

*L468 "... is contributing towards the negative decadal trend." Negative decadal trend in what?*

A negative trend in runoff. We will add this in the revised manuscript.

**Comment 13**

*L469 "... and linked to the summer months" This phrase is vague, and could be made much more specific. What happened in summer? Is it a cause or effect of the drier conditions?*

The trend in runoff was found to be strongest in the summer months. We will rephrase this to clarify.

The updated part:

505     The reconstruction product revealed interesting patterns in long-term runoff trends that align with prior knowledge. Having additional reconstruction for the 1960s and 1970s, it seems that increases in the frequency, rather than in the amplitude of dry years, as well as a decrease in the frequency of wet years, is contributing towards the negative decadal runoff trend. We diagnosed a trend towards lower runoff at the national scale, which was mainly linked to the summer months, where the spatial patterns of runoff indicated increasingly dry conditions particularly in mid-to-high altitudes. We encourage the in-depth

510     investigation of the identified patterns in subsequent studies.

The remaining minor comments will all be adopted in the revised manuscript.

Comment 14

*Table A1 caption "onse" should be "ones"*

Comment 15

*L486 deontes should be denotes*

Comment 16

*L488 temporal_dropoput should be temporal_dropout*

Comment 17

*Table A2 caption. The meanings of the columns "allbasins sqrttrans static" are not defined; please refer the reader to the relevant material in section 3*